# Incremental Sequence Classification
# with Temporal Consistency

**Lucas Maystre**[*]
UiPath
London, UK

**Gabriel Barello**
UiPath
Bellevue, WA, USA

**Tudor Berariu**
UiPath
London, UK

**Aleix Cambray**
UiPath
London, UK

**Rares Dolga**
UiPath & UCL
London, UK

**Alvaro Ortega Gonzalez**
UiPath
London, UK

**Andrei Nica**
UiPath
London, UK

**David Barber**
UiPath & UCL
London, UK

## Abstract

We address the problem of incremental sequence classification, where predictions are updated as new elements in the sequence are revealed. Drawing on temporal-difference learning from reinforcement learning, we identify a temporal-consistency condition that successive predictions should satisfy. We leverage this condition to develop a novel loss function for training incremental sequence classifiers. Through a concrete example, we demonstrate that optimizing this loss can offer substantial gains in data efficiency. We apply our method to text classification tasks and show that it improves predictive accuracy over competing approaches on several benchmark datasets. We further evaluate our approach on the task of verifying large language model generations for correctness in grade-school math problems. Our results show that models trained with our method are better able to distinguish promising generations from unpromising ones after observing only a few tokens.

## 1 Introduction

Learning to classify a sequence $\boldsymbol{x} = (x_1, \ldots, x_T)$ into one of $K$ classes is a fundamental problem in machine learning [44, 16]. In this work, we focus on an incremental variant in which the sequence is revealed element by element, and a predictive model must provide predictions at every timestep. As a concrete example, consider a movie review, represented as a sequence of tokens.

$$\underbrace{\text{A}}_{x_1} \underbrace{\text{touching}}_{x_2} \underbrace{\text{movie}}_{x_3} \underbrace{.}_{x_4} \underbrace{\text{It}}_{x_5} \underbrace{\text{is}}_{x_6} \underbrace{\text{full}}_{x_7} \underbrace{\text{of}}_{x_8} \underbrace{\text{emotions}}_{x_9} \underbrace{\text{and}}_{x_{10}} \underbrace{\text{wonderful}}_{x_{11}} \underbrace{\text{acting}}_{x_{12}} \underbrace{.}_{x_{13}}$$

We ask the question: Can we predict the sentiment of this review using only the first two or three tokens? More generally, can we learn a predictive model that accurately classifies any prefix $\boldsymbol{x}_{\leq t}$, consisting of the first $t \leq T$ elements of a sequence? This problem naturally arises in domains where there is a cost associated to waiting for the full sequence; In healthcare or finance, for example, the cost might be time or opportunity [45, 34]. Recently, sequence classifiers have also been deployed as verifiers to improve applications of large language models (LLMs), where generating full sequences incurs a non-trivial computational cost [8, 39, 31].

A key property of the incremental classification problem is that any calibrated predictive model should be *temporally-consistent*. That is, the predictive class distribution given a prefix $\boldsymbol{x}_{\leq t}$ should equal the expected class distribution given the extended prefix $\boldsymbol{x}_{\leq t+1}$, where the expectation is taken over $x_{t+1}$. We take advantage of this temporal-consistency property to develop a novel loss function for training

---

[*]Corresponding author, e-mail: `lucas@maystre.ch`.

39th Conference on Neural Information Processing Systems (NeurIPS 2025).

incremental classifiers (Section 2). Our approach shares many parallels with temporal-difference (TD) learning [37], an important class of methods in reinforcement learning (RL) [38]. Exploiting these parallels, we study a simple sequence model and demonstrate that optimizing our loss function can lead to substantial data-efficiency gains over the standard maximum-likelihood approach (Section 3).

We apply our method to text classification using decoder-only transformers (Section 4). We first evaluate it on four benchmark datasets and find that models trained with our temporal-consistency loss achieve higher predictive accuracy—both on prefixes and on full sequences. When fine-tuning models of the OPT family [49], the improvement in predictive performance is comparable to increasing the size of the base model by a factor 10. We then explore the emerging application of verifying LLM generations. On GSM8K math word problems [8], our method yields verifiers that accurately distinguish correct from incorrect generations after observing only a few tokens. We illustrate how this enables a more favorable trade-off between answer accuracy and computational cost.

Our approach is inspired by celebrated methods in RL and is rigorously grounded in theory. It is simple to implement, adds negligible computational overhead during training and inference, and improves predictive performance across all the datasets we evaluated. As such, we believe it will be valuable to machine-learning practitioners.

## 1.1 Related work

Early sequence classification [44, 45, 27] addresses a problem that is distinct but complementary to ours: determining *when* enough information has been observed to make a reliable prediction. In contrast, we focus exclusively on *what* to predict at each prefix. Our goal is to maximize classification accuracy across all timesteps, without addressing the decision-making aspect. We note that incremental classifiers trained with our method could serve as components in early classification architectures [34, 4].

TD learning [37, 38] leverages a temporal-consistency condition, closely related to ours, to learn value functions in RL. Classical TD learning is concerned with modeling the reward-to-go, a scalar quantity. Our work can be understood as extending the core idea underpinning TD learning to the multiclass classification setting. Distributional RL [29, 2, 9] shares some algorithmic features, but does not address classification. Cheikhi and Russo [7] recently analyzed the data-efficiency benefits of TD learning, and we build on their results in Section 3. Beyond scalar values, ideas from TD learning have also been applied to survival analysis [26, 40] and early event prediction [48].

Incremental classifiers have recently been applied to verify LLM generations, either during post-training [39, 21] or at inference time [8, 47, 31], where they are referred to as token-level verifiers [8] or outcome-supervised reward models [39, 21]. Verification is typically framed as a binary classification problem, with verifiers trained using a cross-entropy loss against the final observed outcome, a baseline we refer to as *direct cross-entropy* (DCE). Some prior work use soft targets, as we do, but these targets are still derived solely from observed outcomes [41, 20]. A notable exception is Mudgal et al. [31], who frame verification as a regression problem and learn a token-level value function using TD learning. However, they rely on a squared loss, which is ill-suited to binary outcomes [18]. Our work bridges this gap by developing a TD-style approach tailored to classification.

## 2 Direct and temporally-consistent estimators

In order to introduce our approach to learning incremental sequence classifiers, we shift our perspective slightly. Instead of a sequence of arbitrary elements, which we have denoted by $\boldsymbol{x} = (x_1, \ldots, x_T)$ in the introduction, we now consider a sequence of Markov states $\boldsymbol{s} = (s_1, \ldots, s_T)$. That is, we assume that the ground-truth distribution $p(\boldsymbol{s}, y)$ of states and class label satisfies

$$p(s_{t+1} \mid s_t, \ldots, s_1) = p(s_{t+1} \mid s_t), \qquad p(y \mid s_t, \ldots, s_1) = p(y \mid s_t), \qquad (1)$$

for any $t < T$, and for any $t \leq T$ and any $y$, respectively. This is not a restrictive assumption: By slight abuse of notation, one can write $s_t = \boldsymbol{x}_{\leq t}$ and $p(s_{t+1} \mid s_t) = p(x_{t+1} \mid \boldsymbol{x}_{\leq t})$, and trivially satisfy these two Markov properties. The Markov-chain perspective will be useful to relate our developments to temporal-difference learning, and to provide intuition into the statistical benefits of our method in Section 3.

**Notation and problem setup** Let $[M]$ denote the set of consecutive integers $1, \ldots, M$. We are given a dataset of $N$ labelled sequences $\mathcal{D} = \{(\boldsymbol{s}^n, y^n) : n \in [N]\}$, where $y^n \in [K]$, and where different sequences might be of different lengths. We seek to learn a probabilistic classifier $p_{\boldsymbol{\theta}}(y \mid s_t)$, parametrized by $\boldsymbol{\theta}$, that approximates the ground-truth distribution $p(y \mid s_t)$. We denote by $\boldsymbol{p}_{\boldsymbol{\theta}}(\cdot \mid s_t)$ the $K$-dimensional probability vector $[p_{\boldsymbol{\theta}}(1 \mid s_t) \quad \cdots \quad p_{\boldsymbol{\theta}}(K \mid s_t)]$, and by $\boldsymbol{\delta}_y$ the $K$-dimensional one-hot vector with a 1 in position $y$.

**Direct cross-entropy** A natural approach for learning $p_{\boldsymbol{\theta}}(y \mid s_t)$ is to maximize the likelihood of the observed samples from $p(y \mid s_t)$ in the dataset $\mathcal{D}$, or equivalently, to minimize the cross-entropy relative to these samples. Given a labeled sequence $(\boldsymbol{s}, y)$, we define the following loss function, which penalizes deviations from the label $y$ simultaneously across all $t \leq T$.

$$\ell_{\text{DCE}}(\boldsymbol{\theta}; \boldsymbol{s}, y) = -\sum_{t=1}^{T} \log p_{\boldsymbol{\theta}}(y \mid s_t) = \sum_{t=1}^{T} H[\boldsymbol{\delta}_y \,\|\, \boldsymbol{p}_{\boldsymbol{\theta}}(\cdot \mid s_t)], \tag{2}$$

where $H[\boldsymbol{p}\|\boldsymbol{q}] = -\sum_{k=1}^{K} p_k \log q_k$ is the cross-entropy function. Given the full dataset $\mathcal{D}$, we find

$$\boldsymbol{\theta}_{\text{DCE}}^{\star} \leftarrow \arg\min_{\boldsymbol{\theta}} \sum_n \ell_{\text{DCE}}(\boldsymbol{\theta}; \boldsymbol{s}_n, y_n). \tag{3}$$

We call this approach *direct cross-entropy* (DCE), because the target distribution is the observed one-hot class label $\boldsymbol{\delta}_y$. In LLM verification applications, this is the predominant approach to training token-level verifiers [8, 39, 21, 41, 20].

## 2.1 A family of temporally-consistent estimators

Intuitively, a drawback of the direct approach is that, for early states $s_t$ (with $t \ll T$), the training signal provided by observed label $y$ is noisy. Indeed, the prediction $p_{\boldsymbol{\theta}}(y \mid s_t)$ needs to account for two sources of uncertainty, *a*) the uncertainty about how the remainder of the sequence $s_{t+1}, \ldots, s_T$ will unfold, and *b*) the uncertainty about the label $y$ given the last state $s_T$. We make progress by observing that

$$p(y \mid s_t) = \mathbf{E}_{p(s_{t+1}|s_t)}[p(y \mid s_{t+1})], \tag{4}$$

for all $y$ and all $t < T$, an identity that follows from (1). This identity captures a notion of *temporal consistency*: It states that the class distribution at step $t$ is equal to the class distribution at step $t + 1$ on average. Driven by this observation, we propose the following loss function, which for $t < T$ penalizes the temporal inconsistency relative to a reference model parametrized by $\boldsymbol{\theta}'$:

$$\ell_{\text{TC}}(\boldsymbol{\theta}; \boldsymbol{\theta}', \boldsymbol{s}, y) = H[\boldsymbol{\delta}_y \,\|\, \boldsymbol{p}_{\boldsymbol{\theta}}(\cdot \mid s_T)] + \sum_{t=1}^{T-1} H[\boldsymbol{p}_{\boldsymbol{\theta}'}(\cdot \mid s_{t+1}) \,\|\, \boldsymbol{p}_{\boldsymbol{\theta}}(\cdot \mid s_t)]. \tag{5}$$

Comparing (2) and (5) carefully, we can think of this temporal-consistency (TC) loss as replacing the hard targets $\boldsymbol{\delta}_y$ by soft targets $\boldsymbol{p}_{\boldsymbol{\theta}'}(\cdot \mid s_{t+1})$ capturing the predictive distribution at the next state. Given the full dataset $\mathcal{D}$, we start with random parameters $\boldsymbol{\theta}^{(0)}$, and iteratively solve

$$\boldsymbol{\theta}^{(i+1)} \leftarrow \arg\min_{\boldsymbol{\theta}} \sum_n \ell_{\text{TC}}(\boldsymbol{\theta}; \boldsymbol{\theta}^{(i)}, \boldsymbol{s}_n, y_n). \tag{6}$$

In Section 3, we study a tractable setting and show that this iteration converges to a fixed point $\boldsymbol{\theta}_{\text{TC}}^{\star}$, and that the estimator is consistent, i.e., that $p_{\boldsymbol{\theta}_{\text{TC}}^{\star}}(y \mid s_t) \rightarrow p(y \mid s_t)$ as the dataset size $N \rightarrow \infty$.

We can extend the identity (4) to multiple steps, capturing the temporal consistency of class distributions across longer time spans (c.f. Appendix A.1). We use this to formulate a generalized temporal-consistency loss,

$$\begin{aligned} \ell_{\text{TC-}\lambda}(\boldsymbol{\theta}; \boldsymbol{\theta}', \boldsymbol{s}, y) &= \sum_{t=1}^{T} H[\boldsymbol{z}_t \,\|\, \boldsymbol{p}_{\boldsymbol{\theta}}(\cdot \mid s_t)], \\ \boldsymbol{z}_t &= \lambda^{T-t}\boldsymbol{\delta}_y + (1-\lambda)\sum_{k=1}^{T-t} \lambda^{k-1}\boldsymbol{p}_{\boldsymbol{\theta}'}(\cdot \mid s_{t+k}), \end{aligned} \tag{7}$$

where $\lambda \in [0, 1]$ is a hyperparameter. In this loss function, the target $\boldsymbol{z}_t$ providing the training signal is a weighted average of the predictive distributions $k$ steps ahead, with exponentially decreasing weights. The larger $\lambda$ is, the larger the influence of distant states is.[1] In fact, TC-$\lambda$ generalizes both TC (5) and DCE (2). Setting $\lambda = 0$ recovers TC, while $\lambda = 1$ yields DCE. Throughout the paper, we refer to any model trained with the TC-$\lambda$ loss with $\lambda < 1$ as *temporally consistent*.

---

[1]We can think of the mean of the geometric distribution $\lambda/(1 - \lambda)$ as the effective lookahead, i.e., as "how far" the target looks ahead on average.

## 2.2 Connections to temporal-difference learning

Our approach is inspired by temporal-difference (TD) learning, a key idea in reinforcement learning [37, 38, 28]. Quoting Sutton's seminal paper, "whereas conventional prediction-learning methods assign credit by means of the difference between predicted and actual outcomes, the new methods assign credit by means of the difference between temporally successive predictions." Recasting our developments into the language of reinforcement learning, we can think of the probabilistic classifier we learn as a state-value function, capturing the eventual final outcome of a trajectory at any given intermediate state. The temporal-consistency condition (4) is a form of Bellman equation, relating the value function across successive states. Similarly to TD learning, our approach uses the temporal inconsistency of predictions across successive states as the learning signal. Our generalized loss function (7) is inspired by the TD($\lambda$) family of algorithms.

A key difference is that we consider categorical outcomes and a cross-entropy loss, as opposed to the scalar outcomes and squared loss usually employed in TD learning algorithms. In Section 4, we compare our approach to classical value-estimation methods from RL, by treating our $K$-way classification problem as $K$ separate value estimation problems.

## 3 Convergence, consistency and data efficiency

In this section, we analyze the DCE and TC estimators in problems with a finite number of states. We consider tabular models, where a separate probability is learned for each state-class pair. This tractable setting enables a theoretical comparison of the properties of DCE and TC. Specifically, we show that *a)* the TC optimization procedure in (6) converges and *b)* yields a consistent estimator of the true class probabilities, and that *c)* the TC estimator is more data-efficient than DCE. Complete proofs of all propositions are provided in Appendix A.2. Our perspective in this section is partly inspired by dynamic programming and its application to stochastic shortest path problems, as presented in Bertsekas and Tsitsiklis [3, Sec. 2.2].

In the finite-state case, we formalize the problem of multiclass sequence classification as that of finding the absorption probabilities of a Markov chain on $M$ transient states and $K$ absorbing states [32]. The transient states correspond to $M$ distinct values each element of the sequence $\boldsymbol{s}$ can take, and the absorbing states correspond to $K$ classes. The Markov chain is fully characterized by the pair $(\boldsymbol{Q}, \boldsymbol{R})$, where the $M \times M$ matrix $\boldsymbol{Q}$ describes the transition probabilities between every pair of transient states, and the $M \times K$ matrix $\boldsymbol{R}$ describes the transition probabilities from transient to absorbing states. We assume that it is possible to reach a terminal state from any transient state within a finite number of steps. That is, for any transient state $m$, there is a $t \geq 0$ such that $[\boldsymbol{Q}^t \boldsymbol{R}]_{mk} > 0$ for some $k$. Our goal is to estimate the absorption probabilities $p^\star_{mk} = p(y = k \mid s_t = m)$ from data, organized into the $M \times K$ matrix $\boldsymbol{P}^\star$.

Before addressing the estimation problem, note that if the ground-truth transition matrices $\boldsymbol{Q}$ and $\boldsymbol{R}$ are known, $\boldsymbol{P}^\star$ can be computed by starting from an initial guess $\boldsymbol{P}_0$ and refining it iteratively as

$$\boldsymbol{P}_{i+1} = \boldsymbol{Q} \boldsymbol{P}_i + \boldsymbol{R}. \tag{8}$$

**Proposition 1.** *For any $M \times K$ row-stochastic $\boldsymbol{P}_0$, the fixed-point iteration* (8) *converges to $\boldsymbol{P}^\star$.*

Now consider the setting where, instead of access to $\boldsymbol{Q}$ and $\boldsymbol{R}$, we are given a dataset $\mathcal{D} = \{(\boldsymbol{s}^n, y^n) : n \in [N]\}$ of trajectories sampled from the Markov chain. Each trajectory is composed of a transient sequence $\boldsymbol{s}^n$ and terminates in an absorbing state $y^n$. We explore two approaches to estimating the absorption probabilities.

**Direct estimation**   We can estimate the absorption probabilities directly, as the empirical fraction of sequences passing through $m$ ending in $k$. Denoting by $T(\boldsymbol{s})$ the length of sequence $\boldsymbol{s}$ and letting $\mathcal{D}' = \cup_{(\boldsymbol{s}, y) \in \mathcal{D}} \cup_{t \leq T(\boldsymbol{s})} \{(s_t, y)\}$ be the union of all pairs of transient state and eventual absorbing state in $\mathcal{D}$, we have

$$\hat{p}^{\text{dir}}_{mk} = \mathbf{E}_{(s, y) \sim \mathcal{D}'}[\mathbf{1}_{\{y=k\}} \mid s = m],$$

where $(s, y) \sim \mathcal{D}'$ denotes uniform sampling on $\mathcal{D}'$. We collect these estimates into the matrix $\hat{\boldsymbol{P}}^{\text{dir}}$.

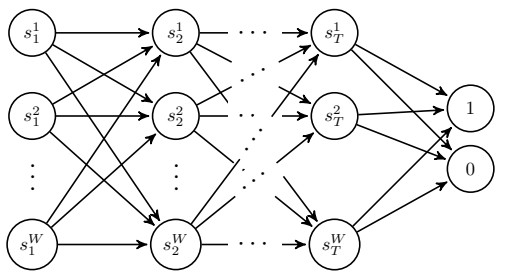 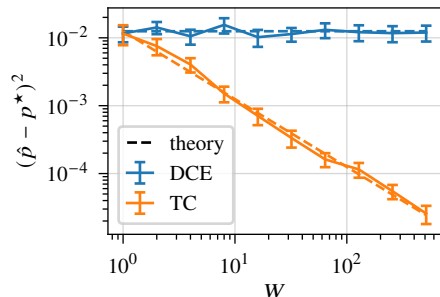

Figure 1: *Left*: Markov chain with $T$ layers of $W$ states each, and two absorbing states. *Right*: Mean-squared error of the direct (DCE) and indirect (TC) estimates for a state in the first layer state as a function of $W$. We set $N = 20W$ and report the mean and $95\%$ confidence intervals over 100 runs.

**Indirect estimation** Alternatively, we can first estimate the matrices $\boldsymbol{Q}$ and $\boldsymbol{R}$ by using the empirical one-hop transition counts. Letting $\mathcal{A} = \cup_{(\boldsymbol{s},y)\in\mathcal{D}}\cup_{t<T(s)}\{(s_t, s_{t+1})\}$ and $\mathcal{B} = \cup_{(\boldsymbol{s},y)\in\mathcal{D}}\{(s_{T(\boldsymbol{s})}, y)\}$ be the union of all transitions between successive transient states and between transient and absorbing states, respectively, and $\mathcal{T} = \mathcal{A} \cup \mathcal{B}$, we write

$$\hat{q}_{mm'} = \mathbf{E}_{(s,s')\sim\mathcal{T}}[\mathbf{1}_{\{s'=m'\}} \mid s = m], \qquad \hat{r}_{mk} = \mathbf{E}_{(s,s')\sim\mathcal{T}}[\mathbf{1}_{\{s'=k\}} \mid s = m]. \qquad (9)$$

We can then plug the estimates $\hat{\boldsymbol{Q}}$ and $\hat{\boldsymbol{R}}$ into the fixed-point iteration (8), and iterate until convergence. We denote the resulting estimate by $\hat{\boldsymbol{P}}^{\text{ind}}$. By Proposition 1, this fixed point corresponds to the exact absorption probability matrix of the empirical Markov chain $(\hat{\boldsymbol{Q}}, \hat{\boldsymbol{R}})$, but it is only an approximation of the ground-truth the absorption probabilities.

The following proposition relates the TC loss (5) and its iterative optimization procedure (6), introduced in Section 2, to the indirect estimator $\hat{\boldsymbol{P}}^{\text{ind}}$ we have just derived.

**Proposition 2.** *Let $p_{\boldsymbol{\theta}}(y = k \mid s_t = m) \doteq \theta_{mk}$. Then, the TC iterative optimization procedure (5) is equivalent to the fixed-point iteration (8) on the empirical Markov chain $(\hat{\boldsymbol{Q}}, \hat{\boldsymbol{R}})$ defined by (9).*

It follows that, in the tabular case, the TC estimator (6) is guaranteed to converge. Furthermore, under mild assumptions on the data-generating distribution[2], the TC estimator is consistent, i.e.,

$$\hat{\boldsymbol{P}}^{\text{ind}} \to \boldsymbol{P}^{\star} \text{ as } N \to \infty.$$

Similarly, we can relate the optimization of the DCE loss (2) to the direct estimator $\hat{\boldsymbol{P}}^{\text{dir}}$ (see Proposition 4 in Appendix A).

### 3.1 Statistical benefits

In general, DCE and TC will not result in the same estimate, i.e., $\hat{\boldsymbol{P}}^{\text{ind}} \neq \hat{\boldsymbol{P}}^{\text{dir}}$, raising the question: Which estimator is better? In related work, Cheikhi and Russo [7] study direct and indirect estimators for the state-value function of a Markov reward process. They find that, depending on the structure of the Markov reward process, the indirect estimator can be significantly more data-efficient than (and is always at least as efficient as) the direct estimator.

In Figure 1, we revisit one of their examples and adapt it to our classification setting. We consider an absorbing Markov chain with $T$ layers of $W$ distinct states each. For any $t < T$, the probability of transitioning from a state at layer $t$ to any other state at layer $t + 1$ is $1/W$. From any state at layer $T$, we transition to one of two absorbing states, 0 and 1, with probability $1/2$ each. Given this, it is easy to verify that $p_{mk}^{\star} = 1/2$ for any transient state $m$ and any absorbing state $k \in \{0, 1\}$. We collect $N$ trajectories of length $T + 1$, sampling the first state uniformly at random, and subsequent states as described previously. The next proposition shows that, for any state in the first layer, the expected squared error of the indirect estimator is $W$ times smaller than that of the direct estimator.

---

[2]Assuming that the initial state distribution and the transition probabilities are such that, for every transient state, there is a non-zero probability of it being sampled in the sequence.

**Proposition 3** (Adapted from [7]). *For any $T \geq 1$ and any state $m$ in the first layer,*

$$\mathbf{E}\left[(\hat{p}_{mk}^{\text{ind}} - p_{mk}^\star)^2\right] \Big/ \mathbf{E}\left[(\hat{p}_{mk}^{\text{dir}} - p_{mk}^\star)^2\right] \xrightarrow{N \to \infty} 1/W.$$

We validate this result empirically in Figure 1 (right), where we report on numerical simulations using $N = 20W$. In this Markov chain, the indirect estimator is increasingly more data-efficient as $W$ increases. Intuitively, the indirect approach acts as a form of regularization, by requiring the solution to satisfy the temporal-consistency property (4), which arises from the problem's Markov structure. The indirect approach benefits from a form of data pooling: By minimizing the inconsistency across successive states instead of regressing the target outcome directly, we take advantage of information from other trajectories that originated from different states and crossed paths.

**Beyond the tabular setting**   In the remainder of this paper, we move beyond the tabular setting studied above and fine-tune parametric large language models on very large state spaces. The theoretical results developed in this section no longer strictly apply, yet we believe that the underlying intuition remains valuable for interpreting our method's performance on complex, real-world tasks. In text classification, for example, many distinct word sequences can result in a similar *semantic state* predictive of the target label. Optimizing for consistency across successive states can therefore improve data efficiency, similarly to the tabular toy example. With temporal consistency, the model implicitly exploits information from sequences that begin differently but reach similar intermediate semantic states, the same data-pooling phenomenon underpinning the gains of the indirect method in Figure 1.

## 4   Empirical evaluation

In this section, we apply our methodology to text classification with decoder-only transformers. First, in Section 4.1, we evaluate multiple different approaches to training incremental classifiers. We compare the predictive performance of models on four well-known text classification benchmarks. Then, in Section 4.2, we consider a concrete application to verifying LLM generations. We show that an accurate token-level correctness classifier enables solving grade-school math problems computationally more efficiently.

**Model architecture & training**   Decoder-only (i.e., causal) transformers [23, 33] are particularly well-suited to incremental sequence classification. As the $t$th output of a decoder on an input sequence $\boldsymbol{x}$ depends only on the prefix $\boldsymbol{x}_{\leq t}$, we can compute predictions at every prefix efficiently, with a single forward inference pass. Throughout this section, we model class probabilities as

$$\boldsymbol{p_\theta}(\cdot \mid \boldsymbol{x}_{\leq t}) = \text{softmax}(\boldsymbol{A}\boldsymbol{h}_t + \boldsymbol{b}), \tag{10}$$

where $\boldsymbol{h}_t \in \mathbf{R}^D$ is the hidden vector at the last layer of a transformer at position $t$, and $\boldsymbol{A} \in \mathbf{R}^{K \times D}$ and $\boldsymbol{b} \in \mathbf{R}^K$ are the parameters of a classification head. We start with a pre-trained language model, and jointly optimize $(\boldsymbol{A}, \boldsymbol{b})$ as well as all the parameters of the transformer, which we collectively refer to as $\boldsymbol{\theta}$. We make two minor practical adjustments with respect to the optimization procedure outlined in Section 2. First, we update the parameters using stochastic gradient updates. Second, we average the loss over all prefixes of each sequence, instead of summing them. This means that every sequence contributes to the loss equally, irrespective of its length. Appendix B describes the precise training procedure and documents how we select hyperparameters.

### 4.1   Text classification datasets

We consider four text classification datasets, spanning tasks such as movie review sentiment prediction (IMDB [25]) and topic classification (OHSUMED [30], NEWSGROUPS [19], AG-NEWS [10]). The number of classes $K$ ranges from 2 to 23. We provide summary statistics for all datasets in Appendix B.2, including the number of training and test samples and the distribution of document length. In addition to the standard setting, where the goal is to predict the class label given the full sequence $\boldsymbol{x}$, we are interested in evaluating the performance of classifiers in the *incremental* setting, where we need to make a prediction after observing only a prefix $\boldsymbol{x}_{\leq t}$ consisting of the first $t$ tokens.

We fine-tune pre-trained language models from the OPT family [49]. Unless otherwise noted, we report results on the $125\,\mathrm{M}$-parameter version of the family, with hidden size $D = 768$. Our primary

Table 1: Predictive performance of incremental text classifiers on four datasets. We report the classification accuracy on 4-token and 16-token prefixes, and on full sequences (all tokens). We highlight the **best** and second-best performing models.

| | OHSUMED | | | NEWSGROUPS | | | IMDB | | | AG-NEWS | | |
|---|---|---|---|---|---|---|---|---|---|---|---|---|
| | 4 | 16 | all | 4 | 16 | all | 4 | 16 | all | 4 | 16 | all |
| Most frequent | 16.0 | 16.0 | 16.0 | 5.3 | 5.3 | 5.3 | 50.0 | 50.0 | 50.0 | 25.0 | 25.0 | 25.0 |
| GPT 4o | 31.5 | 54.0 | 57.5 | 7.5 | 11.0 | 80.4 | 58.0 | 67.0 | 94.3 | 77.4 | 87.4 | 88.3 |
| Filtering | 22.1 | 46.9 | 46.2 | 10.4 | 19.8 | 72.0 | 64.1 | 73.4 | 92.1 | 77.9 | 86.6 | 87.2 |
| Last token | 16.7 | 45.0 | 80.6 | 6.5 | 9.4 | 87.9 | 56.6 | 68.4 | 94.7 | 54.4 | 78.3 | 94.8 |
| Specialist | 24.5 | 59.8 | 80.6 | 23.6 | **47.3** | 87.9 | 59.8 | 73.3 | 94.7 | 77.8 | 91.8 | 94.8 |
| Direct $\ell_2$ loss | 30.3 | 65.0 | 80.4 | 19.7 | 29.9 | 88.3 | 63.6 | 74.7 | 94.3 | 80.0 | 92.6 | 94.7 |
| LSTD($\lambda$) | 32.7 | 64.9 | 78.0 | 26.2 | 36.7 | 87.8 | 64.6 | 75.4 | 94.7 | 81.1 | 92.8 | 94.9 |
| DCE | 30.5 | 65.5 | 81.1 | 27.7 | 40.1 | **89.0** | 63.5 | 74.7 | 94.4 | 80.0 | 92.6 | 94.8 |
| TC-$\lambda$ (ours) | **33.7** | **68.3** | **81.8** | **33.4** | 44.7 | 88.5 | **64.7** | **75.7** | **94.9** | **81.4** | **93.0** | **95.0** |

focus is on comparing models trained by using the DCE (2) and TC-$\lambda$ (7) loss functions. Both of these approaches train a single model to classify prefixes of any length (including the full sequence). In addition, we also consider the following baselines and competing methods.

**Most frequent** A naive baseline that always predicts the most frequent class in the training split.

**GPT-4o** We design a simple prompt asking GPT-4o (version 2024-08-06) to classify a text, by giving the list of classes, one example for each class, and the text itself. We access GPT-4o through OpenAI's commercial API. Details are provided in Appendix B.2.

**Filtering** We fine-tune a class-conditional language model $p_{\theta}(\boldsymbol{x} \mid y)$ on the training data, by using a standard next-token prediction task. At test time, we use Bayes' rule to reverse the conditional probability as $\hat{p}(y \mid x) = Z^{-1} p_{\theta}(\boldsymbol{x} \mid y) p(y)$, where $p(y)$ is a prior distribution and $Z = \sum_y p_{\theta}(\boldsymbol{x} \mid y) p(y)$.

**Last token** Similarly to the DCE loss (2), we minimize the cross-entropy relative to the observed label. But unlike the DCE loss, we include only a single cross-entropy term per sequence, corresponding to the prediction based on the hidden vector $\boldsymbol{h}_T$ at the last token (i.e., capturing the full sequence). To the best of our knowledge, this is the most widespread approach to text classification with decoder-only transformers [43, 12, 17].

**Specialist** This approach is similar to the *last token* method, but improves upon it by training a distinct, specialized model for each prefix length. Instead of full sequences, each model is trained exclusively on prefixes of the corresponding length.

**Direct $\ell_2$ loss** This approach is similar to DCE but removes the softmax and replaces the cross-entropy loss with a squared loss. This is analogous to standard offline Monte Carlo methods for value function estimation in RL [38, Chap. 5].

**LSTD($\lambda$)** The offline temporal-difference value estimation method of Bradtke and Barto [5], recently applied to language modelling in Mudgal et al. [31]. This approach is similar to TC-$\lambda$, without the softmax and with a squared loss instead of the cross-entropy loss.

For TC-$\lambda$ and LSTD($\lambda$), we treat $\lambda$ as a hyperparameter. Values for this and for all other hyperparameters are presented in Appendix B.2. The time it takes to train a TC-$\lambda$ model is virtually indistinguishable (within 1%) from that used to train a DCE model, confirming that the overhead required to compute the soft targets $\{\boldsymbol{z}_t\}$ in (7) is negligible compared to the cost of LLM forward and backward passes.

In Table 1, we report the predictive accuracy of classifiers on hold-out sequences, given prefixes of 4 tokens, 16 tokens, and full sequences. For DCE and TC-$\lambda$ and the corresponding squared-loss methods, we report additional metrics and more prefix lengths in the appendix. As expected, the accuracy of every classifier increases as more tokens are available. However, even with only 4 or 16 tokens (often representing a small fraction of the full sequence), some approaches reach a non-trivial accuracy.

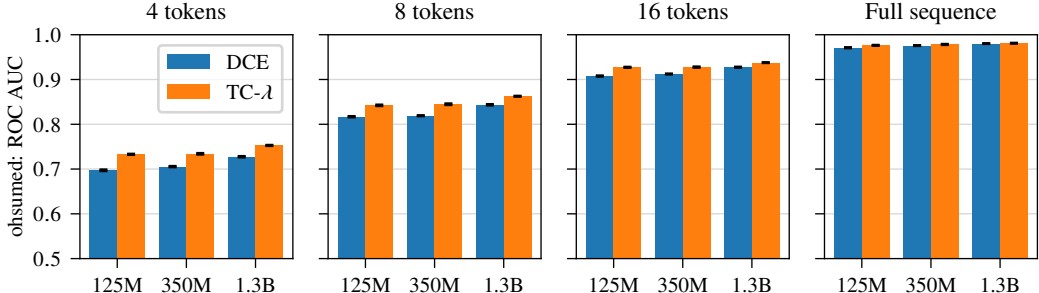

Figure 2: Predictive performance of OPT models with $125\,\text{M}$, $350\,\text{M}$, and $1.3\,\text{B}$ parameters, respectively, on the OHSUMED dataset. We report the area under the ROC curve (mean and $95\%$ confidence interval over 10 runs; higher is better).

We observe that TC-$\lambda$ outperforms DCE and other approaches in all but two cases, where it ranks second. This supports the insights from Section 3.1: incorporating temporal-consistency into the loss function improves predictive performance, even on real-world sequences and with large parametric models. Gains are more substantial for short prefixes, but—perhaps surprisingly—optimizing for temporal-consistency also improves full-sequence classification accuracy. Relatedly, we observe that training incremental classifiers is beneficial even when the goal is only to classify full sequences. Indeed, both DCE and TC-$\lambda$ outperform the *last-token* approach in this setting. This was noticed by Cobbe et al. [8], who suggest that including prefixes in the loss provides a useful auxiliary signal. Our findings reinforce this observation. Finally, we note that the *filtering* method generally underperforms approaches using a classification head, consistent with conventional wisdom that, for classification problems, discriminative models can be more effective than generative models [1].

**Cross-entropy vs. squared loss**  When comparing DCE and TC-$\lambda$, which use a cross-entropy loss, to their squared-loss counterparts (direct $\ell_2$ loss and LSTD($\lambda$), respectively), we observe the following. Temporal consistency tends to improves performance regardless of the loss function, especially on partial sequences. Models trained with a cross-entropy loss perform significantly better than those trained with a squared loss on datasets with many classes (OHSUMED and NEWSGROUPS). For binary or three-way classification tasks (IMDB and AG-NEWS), both loss functions yield a similar accuracy. However, Figure 6 in the appendix shows that methods optimizing a squared loss produce noticeably less well-calibrated predictive probabilities.

**Increasing model size**  Focusing on the OHSUMED dataset and the DCE and TC-$\lambda$ methods, we train OPT models with $125\,\text{M}$, $350\,\text{M}$, and $1.3\,\text{B}$ parameters. Figure 2 shows the area under the ROC curve (ROC AUC) for predictions after 4, 8, and 16 tokens, as well as for full sequences. These results offer the following perspective on the benefits of temporal consistency: DCE requires a model that is approximately $10\times$ larger to match the performance of TC-$\lambda$.

**Varying the temporal-consistency parameter**  In Figure 3 (*left*), we show the accuracy of TC-$\lambda$ models trained on OHSUMED with different values of $\lambda$. The setting $\lambda = 1$, corresponding to DCE, is never optimal, but the best setting depends on the prefix length. When optimizing for full-sequence accuracy, we observe empirically (across the four datasets) that performance is maximized with an effective lookahead of 5–50 tokens, corresponding values of $\lambda$ between 0.8 and 0.98.

**Beyond predictive performance**  We examine the effects of optimizing for temporal consistency more closely. Given predictive distributions $\boldsymbol{p}_t$ and $\boldsymbol{p}_{t+1}$ produced after seeing $t$ and $t + 1$ tokens, respectively, we compute the divergence $\text{KL}[\boldsymbol{p}_{t+1}\|\boldsymbol{p}_t]$. Figure 3 (*right*) shows this divergence, averaged of all successive prefixes of all sequences in the test set. This is the quantity that TC-$\lambda$ explicitly optimizes[3], and unsurprisingly DCE models are significantly less consistent. While we view temporal consistency primarily as a means to improve predictive performance, stable and consistent predictions might be valuable in their own right, for example in decision-making systems [34].

---

[3]$\text{KL}[\boldsymbol{p}_{t+1}\|\boldsymbol{p}_t] = H[\boldsymbol{p}_{t+1}\|\boldsymbol{p}_t] + \text{cst}$, where the constant is independent of $\boldsymbol{p}_t$.

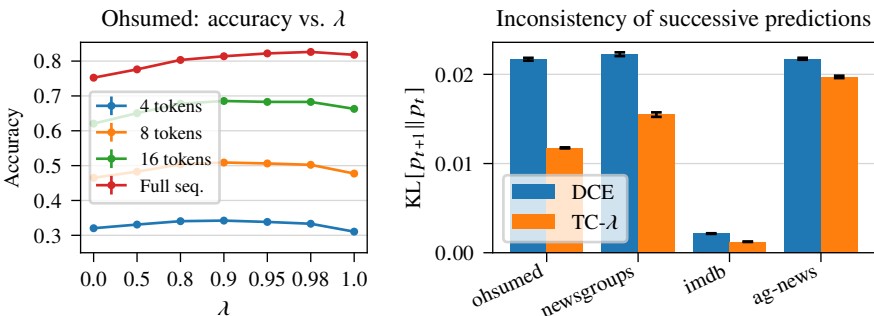

Figure 3: *Left*: Accuracy of OPT-125M classifiers on OHSUMED as a function of $\lambda$ (mean and $95\%$ CI over 5 runs). *Right*: Average KL-divergence between successive predictive distributions (mean and $95\%$ CI over 10 runs). Lower values correspond to predictive distributions that are more similar across successive time steps.

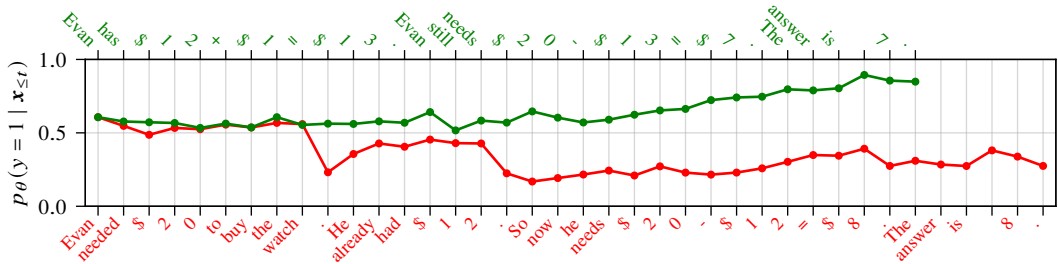

Figure 4: Predicted probability of correctness for two generations of Qwen2.5-0.5B for the prompt *David found $12 on the street. He then gave it to his friend Evan who has $1 and needed to buy a watch worth $20. How much does Evan still need?*

## 4.2 Language model verification on GSM8K

Next, we consider the problem of using an LLM to solve math word problems. In seminal work, Cobbe et al. [8] propose a simple two-step procedure, consisting of *a*) sampling $N$ generations from the LLM, and *b*) scoring them with a *verifier* model, ultimately selecting the generation with the largest score. This best-of-$N$ approach was shown to significantly increase performance over a single generation, at the expense of a larger computational cost (due to sampling $N$ generations). In this setting, *incremental* verifiers bring substantial benefits: If the verifier is able to accurately distinguish between correct and incorrect generations early on, we can focus computational resources on extending only the most promising generations. This idea has recently received significant attention [47, 39, 21, 41, 31, 20]. In this section, we demonstrate that our TC-$\lambda$ approach holds promise for learning better incremental LLM verifiers.

We study GSM8K, a dataset of grade-school math problems and their solutions [8]. For our experiments, we use Qwen2.5-0.5B, a pre-trained language model with $0.5\,B$ parameters that is known to perform well on GSM8K for its size [46]. We proceed as follows. For each of the $7473$ problems in the training set, we sample $32$ generations with temperature $0.7$. We label each generation based on whether or not the answer extracted from the generation matches the ground-truth solution. For each problem, we keep exactly one correct and one incorrect generation (sampled uniformly at random), resulting in a balanced dataset. We then train incremental verifiers by fine-tuning Qwen2.5-0.5B with a classification head, as in (10), using DCE and TC-$\lambda$ (details provided in Appendix B.3). Figure 4 illustrates our setup: Given a problem, our model predicts the probability, token after token, that a generation will eventually produce the correct answer.

**Predictive performance** For this binary classification task, the ROC AUC metric is particularly relevant, as it reflects the probability that the model correctly ranks a randomly-selected correct generation higher than an incorrect one [15]. Figure 5 (*left*) shows the performance of DCE and TC-$\lambda$

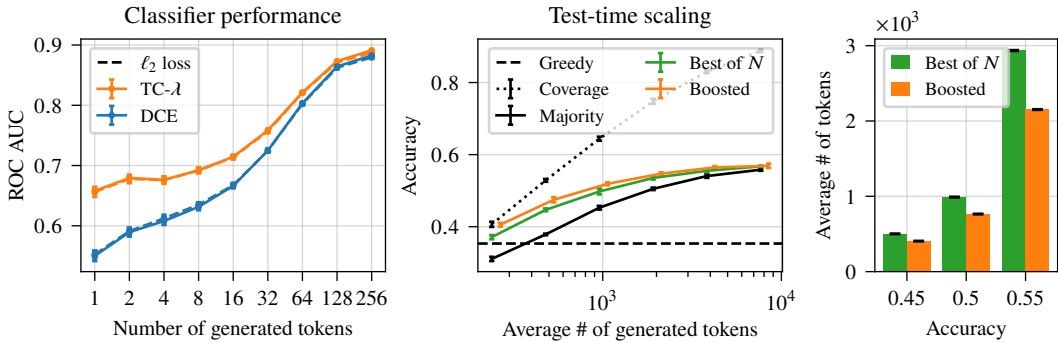

Figure 5: Incremental verification for Qwen2.5-0.5B on GSM8K. *Left*: The TC-$\lambda$ verifier is better at distinguishing between correct and incorrect generations early on. *Center & right*: A better trade-off between accuracy and compute can be obtained by stopping unpromising generations early on.

as a function of the number of generated tokens. We observe that TC-$\lambda$ significantly outperforms DCE when the number of tokens is small. With only 8 tokens, TC-$\lambda$ is almost $70\%$ accurate in distinguishing correct from incorrect generations. Consistent with our findings on the text classification experiments, on this binary classification task, models trained with a squared loss perform almost identically to those trained with a cross-entropy loss in terms of ROC AUC. However, as shown in Figure 8 in the appendix, they produce significantly worse predictive probabilities.

**Application to test-time scaling**   We illustrate the benefit of accurate early correctness predictions in a basic test-time scaling scenario, where we trade off answer accuracy against computational cost, measured by the total number of generated tokens. We propose a simple modification to the best-of-$N$ method, which we call *boosted best-of-$N$* and which is a simplistic version of the speculative rejection method recently proposed by Sun et al. [36]. Sample 10 tokens for each of $2N$ independent generations, rank them using the TC-$\lambda$ incremental verifier, and continue sampling the remaining tokens for the top-$N$ generations until completion. Finally, apply the verifier again to the $N$ completed generations and select the best one. Figure 5 (*center & right*) shows that our approach compares favorably to vanilla best-of-$N$ and to majority voting [42]. For a given level of accuracy, the boosted approach requires $23\%$–$33\%$ fewer tokens.

## 5   Limitations & future work

We have introduced TC-$\lambda$, a loss function for training incremental sequence classifiers that draws on insights from TD learning to improve predictive performance. Our empirical results focus on text classification with transformers, but our approach is architecture-agnostic and applicable to any sequence classification problem. Future work could explore its effectiveness on multimodal applications, such as predicting task success from video frames in robotics [11] and games [13].

Given a limited compute budget, we have prioritized small-scale experiments. This has enabled us to run comprehensive hyperparameter sweeps and to report performance averaged over multiple random seeds, increasing our confidence in the results. Our experiments suggest that TC-$\lambda$ could benefit models at all scales (Figure 2). However, the effectiveness of temporally-consistent methods with models larger than $1.3\,\text{B}$ parameters has not yet been systematically evaluated. Likewise, while our experiments on LLM verification and test-time scaling show promise, further evaluation is needed, particularly in combination with state-of-the-art approaches such as speculative rejection [36].

Finally, a promising direction for future work is applying TC-$\lambda$ to multi-token prediction [35], which has been shown to improve LLM pre-training [14, 22] and accelerate inference [6]. Importantly, calibrated multi-token predictive distributions should satisfy a temporal-consistency condition that is similar to (4), making this a natural fit for our approach.

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

# A    Additional methodological details & proofs

In this appendix, we provide additional details relating to the material presented in Sections 2 and 3.

## A.1    Generalized temporal-consistency condition

The temporal-consistency condition (4) can be extended to $k$-step transitions as follows. Slightly abusing notation and setting $s_{T+1} \equiv \cdots \equiv s_{T+k} \equiv y$, and $p(y \mid s_t) = \mathbf{1}_{\{y=s_t\}}$ for $t > T$, we have

$$p(y \mid s_t) = \mathbf{E}_{p(s_{t+k}|s_t)}[p(y \mid s_{t+k})],$$

for any $y$, any $t$ and any $k$. This follows from the Markov properties (1).

## A.2    Theoretical results

This sections provides proofs for Propositions 1, 2, and 3. It also introduces an additional result, Proposition 4, that is the analogue of Proposition 2 for DCE.

For an $N$-dimensional vector $\boldsymbol{x}$, we denote the $L_1$-norm as $\|\boldsymbol{x}\|_1 = \sum_{n=1}^{N} |x_i|$. Similarly, for an $N \times M$ matrix $\boldsymbol{X}$, we denote the induced $\infty$-norm as $\|\boldsymbol{X}\|_\infty = \max_{n=1}^{N} \|\boldsymbol{x}_n\|_1$, where $\boldsymbol{x}_n$ is the $n$th row of $\boldsymbol{X}$.

*Proof of Proposition 1.* Denote by $\boldsymbol{a}_m^t$ the $m$th row of the matrix $\boldsymbol{Q}^t \boldsymbol{R}$. By assumption, we know that there is a $\tau \in \mathbf{N}_{\geq 0}$ and a $\varepsilon > 0$ such that $\|\boldsymbol{a}_m^\tau\|_1 \geq \varepsilon$ for all $m$. Let $F : \mathbf{R}^{M \times K} \to \mathbf{R}^{M \times K}$ be the linear operator defined by $F(\boldsymbol{P}) = \boldsymbol{Q}\boldsymbol{P} + \boldsymbol{R}$. We will show that $F^{\tau+1}$ is a contraction mapping in the induced $\infty$-norm, that is,

$$\|F^{\tau+1}(\boldsymbol{P}) - F^{\tau+1}(\boldsymbol{P}')\|_\infty \leq (1 - \varepsilon)\|\boldsymbol{P} - \boldsymbol{P}'\|_\infty,$$

for any two $M \times K$ row-stochastic matrices $\boldsymbol{P}, \boldsymbol{P}'$. By the Banach fixed-point theorem, it follows that $F$ admits a unique fixed point $\boldsymbol{P}^\star = \lim_{t \to \infty} F^t(\boldsymbol{P}_0)$ for any initial $\boldsymbol{P}_0$.

By construction, the matrix $\boldsymbol{U} \doteq [\boldsymbol{Q} \quad \boldsymbol{R}]$ is row-stochastic, and thus $\|\boldsymbol{U}\|_\infty = 1$ and $\|\boldsymbol{Q}\|_\infty \leq 1$. It follows that $\boldsymbol{Q}^\tau \boldsymbol{U} = [\boldsymbol{Q}^{\tau+1} \quad \boldsymbol{Q}^\tau \boldsymbol{R}]$ is such that $\|\boldsymbol{Q}^\tau \boldsymbol{U}\| \leq \|\boldsymbol{Q}\|_\infty^\tau \|\boldsymbol{U}\|_\infty \leq 1$. Since every row of $\boldsymbol{Q}^\tau \boldsymbol{R}$ has $L_1$-norm at least $\varepsilon$, it must be that $\|\boldsymbol{Q}^{\tau+1}\|_\infty \leq 1 - \varepsilon$. It follows that

$$\|F^{\tau+1}(\boldsymbol{P}) - F^{\tau+1}(\boldsymbol{P}')\|_\infty = \|\boldsymbol{Q}^{\tau+1}(\boldsymbol{P} - \boldsymbol{P}')\|_\infty \leq \|\boldsymbol{Q}^{\tau+1}\|_\infty \|\boldsymbol{P} - \boldsymbol{P}'\|_\infty$$
$$\leq (1 - \varepsilon)\|\boldsymbol{P} - \boldsymbol{P}'\|_\infty. \qquad \square$$

*Proof of Proposition 2.* Consider one step of the TC optimization problem (6), where we denote the tabular model by using the $M \times K$ matrix $\boldsymbol{\Theta} = [\boldsymbol{\theta}_m]$. Letting $c_m = \sum_{(s,s') \in \mathcal{T}} \mathbf{1}_{\{s=m\}}$, we have

$$\boldsymbol{\Theta}^{(i+1)} \in \arg\min_{\boldsymbol{\Theta}} \sum_n \ell_{\text{TC}}(\boldsymbol{\Theta}, \boldsymbol{\Theta}^{(i)}, \boldsymbol{s}_n, y_n)$$

$$= \arg\min_{\boldsymbol{\Theta}} \sum_{(s,y) \in \mathcal{B}} H[\boldsymbol{\delta}_y \| \boldsymbol{\theta}_s] + \sum_{(s,s') \in \mathcal{A}} H[\boldsymbol{\theta}_{s'}^{(i)} \| \boldsymbol{\theta}_s]$$

$$= \arg\min_{\boldsymbol{\Theta}} \sum_m c_m \left\{ \sum_k \hat{r}_{mk} H[\boldsymbol{\delta}_k \| \boldsymbol{\theta}_m] + \sum_{m'} \hat{q}_{mm'} H[\boldsymbol{\theta}_{m'}^{(i)} \| \boldsymbol{\theta}_m] \right\}$$

$$= \arg\min_{\boldsymbol{\Theta}} \sum_m c_m H[\hat{\boldsymbol{r}}_m + \hat{\boldsymbol{q}}_m^\top \boldsymbol{\Theta}^{(i)} \| \boldsymbol{\theta}_m],$$

where $\hat{\boldsymbol{r}}_m = [\hat{r}_{mk}] \in \mathbf{R}^K$, and $\hat{\boldsymbol{q}}_m = [\hat{q}_{mm'}] \in \mathbf{R}^M$, and where the last equality uses the linearity of the cross-entropy with respect to the target distribution. It follows that the cross-entropy is minimized if $\boldsymbol{\Theta} = \hat{\boldsymbol{Q}}\boldsymbol{\Theta}^{(i)} + \hat{\boldsymbol{R}}$, which corresponds exactly to (8). $\qquad \square$

*Proof of Proposition 3.* The Markov chain of Figure 1 has exactly two absorbing states, and the absorption probabilities sum up to 1. As such, we can focus on the variance of the estimator for $p_{m1}^\star$. We will construct a Markov reward process whose state-value function $V(m)$ is equivalent to the

absorption probability $p_{m1}^\star$. To this end, we collapse the two absorbing states $0$ and $1$ into a single terminal state denoted by $\varnothing$. We instantiate the reward distribution as follows. For any transition between two transient states, the reward is always $0$. For any transition between a state at layer $T$ and the absorbing state $\varnothing$, the reward is $1$ with probability $1/2$ and zero otherwise.

We can now recast our problem by using the terminology of Cheikhi and Russo [7, Sec. 7]. For every transient state $m$ we have

$$V(m) = p_{m1}^\star, \qquad\qquad V^{\mathrm{MC}}(m) = \hat{p}_{m1}^{\mathrm{dir}}, \qquad\qquad V^{\mathrm{TD}}(m) = \hat{p}_{m1}^{\mathrm{ind}}.$$

We focus on states in the first and second layer. Given the sampling process we have defined in the main text, any given state in the second layer will appear in $N/W$ trajectories in expectation, and the transition between any pair of first and second layer states will appear in $N/W^2$ trajectories in expectation. It follows that, for any state $m$ in the first layer and any state $m'$ in the second layer, the coupling coefficient and the inverse trajectory pooling coefficient are given by

$$C(m, m') = 1/W, \qquad\qquad C(m) = 1/W,$$

respectively. We can then apply Theorem 7.2 in Cheikhi and Russo [7] to obtain the desired result. $\qquad\square$

The next proposition relates the DCE loss (2) and the optimization problem (3), introduced in Section 2, to the direct estimator $\hat{\boldsymbol{P}}^{\mathrm{dir}}$ presented in Section 3.

**Proposition 4.** *Let $p_{\boldsymbol{\theta}}(y = k \mid s_t = m) \doteq \theta_{mk}$. Then, $\hat{\boldsymbol{P}}^{\mathrm{dir}}$ is a solution of the DCE optimization problem* (3).

*Proof.* Denote the tabular model by using the $M \times K$ matrix $\boldsymbol{\Theta} = [\boldsymbol{\theta}_m]$. Letting $c_m = \sum_{(s, y') \in \mathcal{D}'} \mathbf{1}_{\{s = m\}}$, we have

$$\boldsymbol{\Theta}_{\mathrm{DCE}}^\star \in \arg\min_{\boldsymbol{\Theta}} \sum_n \ell_{\mathrm{DCE}}(\boldsymbol{\Theta}, \boldsymbol{s}_n, y_n)$$

$$= \arg\min_{\boldsymbol{\Theta}} \sum_{(s, y) \in \mathcal{D}'} H[\boldsymbol{\delta}_y \| \boldsymbol{\theta}_s]$$

$$= \arg\min_{\boldsymbol{\Theta}} \sum_m c_m \left\{ \sum_k \hat{p}_{mk}^{\mathrm{dir}} H[\boldsymbol{\delta}_k \| \boldsymbol{\theta}_m] \right\}$$

$$= \arg\min_{\boldsymbol{\Theta}} \sum_m c_m H[\hat{\boldsymbol{p}}_m^{\mathrm{dir}} \| \boldsymbol{\theta}_m],$$

where $\hat{\boldsymbol{p}}_m^{\mathrm{dir}} = [\hat{p}_{mk}^{\mathrm{dir}}] \in \mathbf{R}^K$, and where the last equality uses the linearity of the cross-entropy with respect to the target distribution. It follows that the cross-entropy is minimized if $\boldsymbol{\Theta} = \hat{\boldsymbol{P}}^{\mathrm{dir}}$. $\qquad\square$

## B Additional details on experimental evaluation

This section provides additional details on the experiments presented in the main paper. In Section B.1, we describe the precise procedure we employ to train and select models. In Section B.2, we provide more details on the text classificiation experiments of Section 4.1. In Section B.3, we provide more details on the verification experiments of Section 4.2.

### B.1 Training, model selection & metrics

Algorithm 1 presents one step of the training loop for the TC-$\lambda$ and DCE approaches. Note that DCE is obtained simply by setting $\lambda = 1$, as explained in Section 2.1. With respect to the iterative optimization procedure (6), we make two minor practical adjustments. First, we update the parameters using stochastic gradient updates. Second, we average the loss over all prefixes of each sequence, instead of summing them. This means that every sequence contributes to the loss equally, irrespective of its length.

We run our experiments on an `a3-highgpu-8g` instance on Google Cloud, with $208$ vCPUs, $1872$ GB of memory, and $8$ NVIDIA H100 GPUs. We ensure that every experiment runs on a single

**Algorithm 1** TC-$\lambda$ incremental classifier: single training step
___
**Require:** minibatch $\mathcal{B}$, parameters $\boldsymbol{\theta}$, temporal-consistency parameter $\lambda$, learning rate $\eta$
1:   $\ell(\boldsymbol{\theta}) \leftarrow 0$                                                     $\triangleright$ Initialize the loss function.
2:   **for** $(\boldsymbol{x}, y) \in \mathcal{B}$ **do**                                      $\triangleright$ Iterate over the minibatch.
3:       $T \leftarrow \text{length}(\boldsymbol{x})$
4:       $\boldsymbol{z}_T \leftarrow \boldsymbol{\delta}_y$
5:       **for** $t = T-1, \ldots, 1$ **do**
6:           $\boldsymbol{z}_t \leftarrow \lambda \boldsymbol{z}_{t+1} + (1-\lambda) \boldsymbol{p}_{\boldsymbol{\theta}}(\cdot \mid \boldsymbol{x}_{\leq t+1})$
7:       $\ell(\boldsymbol{\theta}) \leftarrow \ell(\boldsymbol{\theta}) + \frac{1}{T}\sum_{t=1}^{T} H[\text{stopgrad}(\boldsymbol{z}_t) \| \boldsymbol{p}_{\boldsymbol{\theta}}(\cdot \mid \boldsymbol{x}_{\leq t})]$
8:   **return** $\boldsymbol{\theta} - \frac{\eta}{|\mathcal{B}|}\boldsymbol{\nabla}_{\boldsymbol{\theta}}\ell(\boldsymbol{\theta})$                            $\triangleright$ Update the parameters.
___

GPU. For every dataset, we set the batch size to maximize GPU utilization. We use the AdamW optimizer [24] with a dynamic learning rate. The learning rate starts at zero, increases linearly over a warmup period, then decreases linearly to zero at the end of the optimization process. We vary the following hyperparameters:

- the number of training epochs,

- the maximum learning rate,

- the warmup period (as a ratio of the total number of training steps),

- the weight decay, and

- the temporal-consistency parameter $\lambda$ (for TC-$\lambda$ only).

We run experiments on a grid of hyperparameter configurations, and we select the configuration that maximizes the full-sequence predictive accuracy on a small dataset of held-out sequences. Finally, throughout section 4, we report the mean and standard deviation of the performance of the winning hyperparameter configuration on the full test set, across 10 training runs with different random seeds.

We consider three metrics to measure the quality of a model $p_{\boldsymbol{\theta}}(y \mid \boldsymbol{x})$ on held-out data. The average accuracy is the fraction of examples where $\arg\max_y p_{\boldsymbol{\theta}}(y \mid \boldsymbol{x})$ matches the ground-truth label $y^\star$ (higher is better). The average negative log-likelihood (NLL) is the empirical average of $-\log p_{\boldsymbol{\theta}}(y^\star \mid \boldsymbol{x})$ (lower is better). The area under the ROC curve (ROC AUC) captures the model's ability to discriminate between a random positive and negative example; In the multiclass setting, we use the one-vs-rest macro-average version. A higher ROC AUC is better.

## B.2   Text classification datasets

We evaluate models on the following four well-known text classification benchmarks. Summary statistics and licensing terms for each dataset are provided in Table 2.

**OHSUMED**   The dataset contains abstracts of publications in medical journals [30]. The task is to categorize each abstract into one of 23 themes, corresponding to sub-categories of cardiovascular diseases. We use the archive `ohsumed-all-docs.tar.gz` available at `https://disi.unitn.it/moschitti/corpora.htm`.

**NEWSGROUPS**   The dataset contains newsgroup documents from 20 different newsgroups [19]. The task is to identify which newsgroup the document comes from. We use the version of the dataset hosted at `https://huggingface.co/datasets/google-research-datasets/newsgroup`.

**IMDB**   The dataset contains movie reviews from the IMDb website [25]. The task consists of identifying the sentiment of the review (positive or negative). We use the version of the dataset hosted at `https://huggingface.co/datasets/stanfordnlp/imdb`.

**AG-NEWS**   The dataset contains short news articles [10]. The task is to identify the topic of each article. We use the version of the dataset hosted at `https://huggingface.co/datasets/fancyzhx/ag_news`.

Table 2: Summary statistics for the text classification datasets. Statistics on the sequence length assume that the text is tokenized with the GPT-2 tokenizer [33].

| Dataset | License | $N_{\text{train}}$ | $N_{\text{test}}$ | $K$ | Seq. length percentiles | | |
|---|---|---|---|---|---|---|---|
| | | | | | 50th | 90th | 99th |
| OHSUMED | CC BY-NC 4.0 | 11 520 | 6782 | 23 | 270 | 430 | 604 |
| NEWSGROUPS | unknown | 11 314 | 7532 | 20 | 373 | 935 | 4226 |
| IMDB | unknown | 25 000 | 25 000 | 2 | 221 | 584 | 1159 |
| AG-NEWS | unknown | 120 000 | 7600 | 4 | 51 | 70 | 122 |

The NEWSGROUPS, IMDB and AG-NEWS datasets are provided with separate train and test splits, which we reuse as-is. For OHSUMED, we create our own train and test splits, by partitioning the data uniformly at random.

We fine-tune pre-trained models from the OPT family [49], which are made publicly available under the OPT-175B license[4]. Table 3 provides the hyperparameter configurations for the fine-tuned OPT-125M models whose performance we report in Table 1 in the main text. These hyperparameters are found using the process outlined in Section B.1.

Figure 6 presents detailed results for DCE, TC-$\lambda$, and the corresponding squared-loss variants, Direct $\ell_2$ loss and LSTD($\lambda$), for prefixes of length $2^i, i = 0, \ldots, 9$, across three metrics. We report the mean and the $95\%$ confidence interval of 10 independent seeds, but the standard error is too small to be visible on the plot. On all datasets, the TC-$\lambda$ models outperform the DCE models on almost all metrics at almost all prefix lengths.

**Compute resources** Each experiment, consisting of training an evaluating a model, takes 5–90 minutes on a single GPU, depending on the dataset, the size of the model, and the number of epochs. We estimate that the total compute used for all the experiments performed in the paper, including the hyperparameter sweeps, amounts to approximately 2000 GPU hours.

### B.2.1 GPT-4o baseline

We present the templates used for prompting GPT-4o in Figure 7. We use the structured outputs API to ensure that the model always returns exactly one valid class label. For cost reasons, we sample of a subset of 200 examples uniformly at random from the test set for each dataset and each prefix length (4, 16, and all tokens), and restrict our evaluation to this subset. Approximately $0.7\%$ of requests trigger the ChatGPT filters (mostly in the IMDB and NEWSGROUPS datasets). We simply omit the corresponding examples from the evaluation.

---

[4]See: `https://github.com/facebookresearch/metaseq/blob/main/projects/OPT/MODEL_LICENSE.md`.

Table 3: Hyperparameters used to fine-tune the OPT-125M models reported in the paper.

| Dataset | Model | Batch size | Epochs | Max. LR | Warmup | Weight decay | $\lambda$ |
|---|---|---|---|---|---|---|---|
| ohsumed | Filtering | 32 | 3 | $2 \times 10^{-4}$ | 0.03 | $1 \times 10^{-3}$ | — |
| | Specialist, 4 | 64 | 2 | $5 \times 10^{-4}$ | 0.10 | $1 \times 10^{-4}$ | — |
| | Specialist, 16 | 64 | 2 | $5 \times 10^{-4}$ | 0.10 | $1 \times 10^{-3}$ | — |
| | Last token | 64 | 4 | $1 \times 10^{-4}$ | 0.10 | $1 \times 10^{-4}$ | — |
| | DCE | 128 | 2 | $2 \times 10^{-4}$ | 0.10 | $1 \times 10^{-4}$ | — |
| | TC-$\lambda$ | 128 | 4 | $1 \times 10^{-4}$ | 0.10 | $1 \times 10^{-3}$ | 0.95 |
| | $\ell_2$ loss, direct | 128 | 2 | $2 \times 10^{-4}$ | 0.10 | $1 \times 10^{-4}$ | — |
| | $\ell_2$ loss, TD-$\lambda$ | 128 | 4 | $1 \times 10^{-4}$ | 0.10 | $1 \times 10^{-3}$ | 0.95 |
| newsgroups | Filtering | 4 | 4 | $5 \times 10^{-5}$ | 0.03 | 0 | — |
| | Specialist, 4 | 64 | 2 | $2 \times 10^{-5}$ | 0.10 | $1 \times 10^{-4}$ | — |
| | Specialist, 16 | 64 | 4 | $5 \times 10^{-5}$ | 0.10 | $1 \times 10^{-2}$ | — |
| | Last token | 64 | 4 | $5 \times 10^{-5}$ | 0.10 | $1 \times 10^{-4}$ | — |
| | DCE | 64 | 4 | $1 \times 10^{-4}$ | 0.03 | $1 \times 10^{-5}$ | — |
| | TC-$\lambda$ | 64 | 4 | $1 \times 10^{-4}$ | 0.10 | $1 \times 10^{-3}$ | 0.98 |
| | $\ell_2$ loss, direct | 64 | 4 | $1 \times 10^{-4}$ | 0.03 | $1 \times 10^{-5}$ | — |
| | $\ell_2$ loss, TD-$\lambda$ | 64 | 4 | $1 \times 10^{-4}$ | 0.10 | $1 \times 10^{-3}$ | 0.98 |
| imdb | Filtering | 8 | 2 | $5 \times 10^{-4}$ | 0.03 | $1 \times 10^{-3}$ | — |
| | Specialist, 4 | 8 | 2 | $5 \times 10^{-5}$ | 0.10 | $1 \times 10^{-4}$ | — |
| | Specialist, 16 | 8 | 2 | $2 \times 10^{-5}$ | 0.10 | $1 \times 10^{-2}$ | — |
| | Last token | 8 | 2 | $2 \times 10^{-5}$ | 0.10 | $1 \times 10^{-4}$ | — |
| | DCE | 8 | 1 | $2 \times 10^{-5}$ | 0.10 | $1 \times 10^{-3}$ | — |
| | TC-$\lambda$ | 8 | 2 | $2 \times 10^{-5}$ | 0.10 | $1 \times 10^{-3}$ | 0.80 |
| | $\ell_2$ loss, direct | 8 | 1 | $2 \times 10^{-5}$ | 0.10 | $1 \times 10^{-3}$ | — |
| | $\ell_2$ loss, TD-$\lambda$ | 8 | 2 | $2 \times 10^{-5}$ | 0.10 | $1 \times 10^{-3}$ | 0.80 |
| ag-news | Filtering | 64 | 2 | $5 \times 10^{-5}$ | 0.03 | 0 | — |
| | Specialist, 4 | 64 | 2 | $1 \times 10^{-4}$ | 0.10 | $1 \times 10^{-2}$ | — |
| | Specialist, 16 | 64 | 2 | $1 \times 10^{-4}$ | 0.10 | $1 \times 10^{-3}$ | — |
| | Last token | 64 | 2 | $1 \times 10^{-4}$ | 0.10 | $1 \times 10^{-4}$ | — |
| | DCE | 128 | 2 | $1 \times 10^{-4}$ | 0.03 | $1 \times 10^{-3}$ | — |
| | TC-$\lambda$ | 128 | 2 | $1 \times 10^{-4}$ | 0.10 | $1 \times 10^{-2}$ | 0.90 |
| | $\ell_2$ loss, direct | 128 | 2 | $1 \times 10^{-4}$ | 0.03 | $1 \times 10^{-3}$ | — |
| | $\ell_2$ loss, TD-$\lambda$ | 128 | 2 | $1 \times 10^{-4}$ | 0.10 | $1 \times 10^{-2}$ | 0.90 |

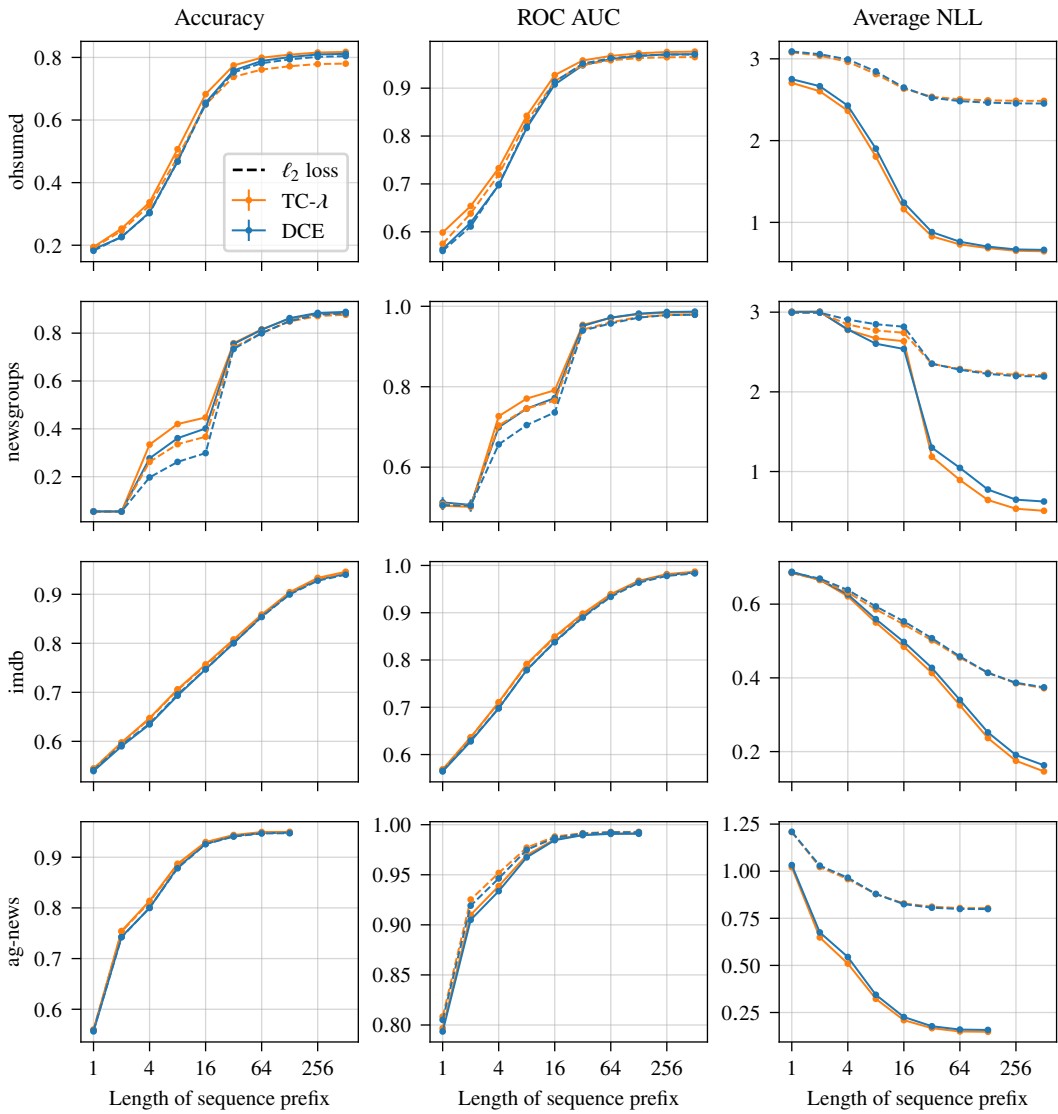

Figure 6: Detailed results for OPT-125M models trained with DCE, TC-$\lambda$, and the corresponding squared-loss variants on the text classification datasets. Accuracy and ROC AUC: higher is better. Average NLL: lower is better. We report means and 95% confidence intervals over 10 runs (too small to be visible).

Listing 1: System prompt

```
You are a helpful AI
assistant
specializing in
classifying text. The
 possible class
labels are:

{class_names}

Here are some
examples of text with
 their corresponding
class labels:

{examples}
```

Listing 2: Single example

```
### BEGIN TEXT ###
{text}
### END TEXT ###
label: {label}
```

Listing 3: User prompt

```
What is the label of
the following text?

### BEGIN TEXT ###
{text}
### END TEXT ###
```

Figure 7: Templates used for prompting GPT-4o.

Table 4: Hyperparameters used to fine-tune the Qwen2.5-0.5B models reported in the paper.

| Dataset | Model | Batch size | Epochs | Max. LR | Warmup | Weight decay | $\lambda$ |
|---------|-------|-----------|--------|---------|--------|--------------|-----------|
| GSM8K | DCE | 48 | 2 | $5 \times 10^{-6}$ | 0.03 | $1 \times 10^{-2}$ | — |
| | TC-$\lambda$ | 48 | 2 | $5 \times 10^{-6}$ | 0.10 | $1 \times 10^{-5}$ | 0.95 |
| | Direct $\ell_2$ loss | 48 | 2 | $5 \times 10^{-6}$ | 0.03 | $1 \times 10^{-2}$ | — |
| | LSTD($\lambda$) | 48 | 2 | $5 \times 10^{-6}$ | 0.10 | $1 \times 10^{-5}$ | 0.95 |

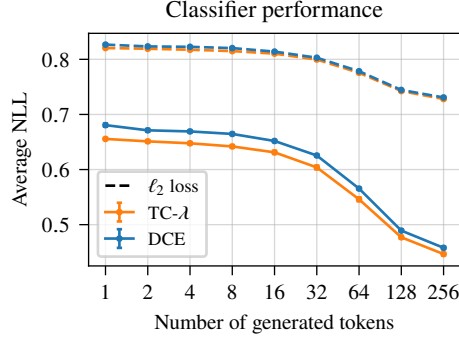

Figure 8: Predictive negative log-likelihood of incremental Qwen2.5-0.5B verifiers on GSM8K (mean and $95\%$ CI over 10 runs).

## B.3 Language model verification on GSM8K

For the language model verification experiments, we use the Qwen2.5-0.5B pre-trained language model [46], which is publicly available under the Apache 2.0 license. The GSM8K dataset [8] is publicly available under the MIT license. Hyperparameter selection follows the procedure outlined for the text classification experiments in Section B.1, with one small change: instead of selecting the best hyperparameter configuration based on the full-sequence accuracy, we select it based on the full-sequence ROC AUC. Table 4 provides the hyperparameter configurations for the fine-tuned Qwen2.5-0.5B models whose performance we report in Figure 5 in the main text.

In these experiments, there is one important conceptual difference with respect to the text classification experiments of Section 4.1. The sequence we seek to classify consists of the *generated* tokens only, but the verifier also needs to access the prompt (i.e., the problem statement). We implement this by concatenating the generated response to the prompt. However, when computing the loss, we mask out the terms that correspond to tokens in the prompt.

**Predictive NLL**  Figure 5 (left) in the main text shows that models trained with a squared loss achieve essentially the same ROC AUC as those trained with cross-entropy. Figure 8 further shows that models trained with a cross-entropy loss achieve substantially lower (i.e., better) negative log-likelihood. Thus, when accurate & calibrated predictive uncertainties are important, optimizing the cross-entropy objective performs better in practice.

