# OpenReview forum: "Incremental Sequence Classification with Temporal Consistency"
_NeurIPS.cc/2025/Conference — NeurIPS 2025 spotlight_

### Official Review · Reviewer_qrJB · 2025-07-02

**Clarity:** 3
**Significance:** 3
**Originality:** 2
**Rating:** 5
**Confidence:** 3

**Summary:**

The authors consider the problem of incremental sequence classification, where the goal is to classify partial sequences at every step that elements of the sequence are revealed. Drawing parallels to TD-learning, which considers a loss to the value at the next time step, as opposed to Monte-Carlo based methods that use full sequence reward, the authors introduce a cross-entropy-like method that minimizes incremental divergences. The authors theoretically analyze their work using a Markov-chain perspective, and empirically compare against standard cross entropy methods in several text classification tasks with decoder-only transformers, demonstrating improvement over the standard cross entropy baseline.

**Questions:**

How does your work compare to Nested Variational Inference https://arxiv.org/abs/2106.11302? Intuitively it feels to me like there's similarity in terms of minimizing divergences to the next-step distribution.

What kinds of confidence intervals did you use?

Figure 5 left - how come 1 token TC outperforms? How is it different from DCE if there's only 1 token? Shouldn't we expect consistency to make more of a difference with more generated tokens? I'm probably missing something here. Is it that you still generate the whole set of tokens, but only evaluate on a cut off set using labels from the whole set?

I think my score is unlikely to change (more clarity would be nice, but I already gave a pretty high score, and I don't see this paper having "groundbreaking impact"). I suppose my score could decrease if discussion reveals weaknesses.

**Ethical Concerns:**

["NO or VERY MINOR ethics concerns only"]

**Final Justification:**

I am keeping my score of accept. Discussion with authors was helpful but did not really change my evaluation.

**Limitations:**

The authors acknowledge the limitations of their theoretical results in not being directly applicable to the language models they investigate. This honesty is good, but it does make me wonder how much value the theory provides. More discussion about how different they expect things to be, along with some further empirical investigation/comparison with the tabular setting or small-scale language modelling settings, might be useful.

Regarding the potential negative societal impact, at least the authors are honest in saying that they don't address this, and they do at least acknowledge that "downstream applications of such models may carry societal implications". And while I do see the reasoning behind downstream application mattering more than the fundamental algorithm, I still think more could have been done (at least in the Appendix and/or checklist) to discuss possible negative impacts (even if unlikely), as fundamental algorithm advancements also enable more downstream applications.

**Paper Formatting Concerns:**

No major formatting issues.

**Quality:**

3

**Strengths And Weaknesses:**

Strengths:

Overall, I think this paper is strong, which is why this review is relatively short.

Overall idea makes sense and is definitely worth exploring (I was actually surprised that this hadn't been done before; I searched online for about 30 minutes and couldn't find it; doesn't mean it hasn't been done before, but I trust that the authors did a much more comprehensive search than I did). Theory makes sense, provides insights, and seems correct to me (though I only carefully checked a few things).

Evaluations are very strong. Comprehensive baselines and ablations are considered, including things I thought of and more. Confidence intervals are included (although unfortunately not on all results). A range of datasets and models are tested. Results appear strong and convincing, supporting the value of the proposed method.

Paper is well written overall and reasonably clear. I would have liked more explanation in many areas, but I understand that space is limited so sometimes there's no getting around the issue of having to write in a terse way.

Weaknesses:

Line 18 has a typo - x8 appears twice.

Line 169 - what are these mild assumptions? Should be stated somewhere (or a reference given).

As the authors acknowledge, applicability of the theory to practical settings may be limited.

Would have liked to see confidence intervals on all results (including e.g., Figure 3 left and Figure 5 left).

For originality, I'm only giving a "fair" rating because I think the main idea is slightly marginal. That said, lots of published conference works are quite marginal, so I don't think this should prevent this paper from being accepted; I definitely think this paper meets the standards for publication.

---

> ### Author Rebuttal · Authors · 2025-07-30
>
> Thank you for your review!
>
> > Line 169 - what are these mild assumptions? Should be stated somewhere (or a reference given)
>
> We assume that the initial state distribution and the transition probabilities are such that, for every transient state, there is a non-zero probability of it being sampled in the sequence. We agree with your suggestion and will clarify this in the next version of the manuscript.
>
> > Would have liked to see confidence intervals on all results (including e.g., Figure 3 left and Figure 5 left).
>
> Figure 3 (left) and Figure 5 (left) do actually include confidence intervals (95% confidence intervals based on the standard error of the mean computed over 5 and 10 random seeds, respectively), but they are so small that they are barely visible. As far as we are aware, Table 1 is the only float where we present experimental results without confidence intervals (although the corresponding confidence intervals can be read off of Figure 6 in the appendix)
>
> We acknowledge that it is not at all easy to see that confidence intervals are present in the figures you mentioned, and we will experiment with alternative presentation styles (we also welcome your suggestions in that regard).
>
> > How does your work compare to Nested Variational Inference https://arxiv.org/abs/2106.11302?
>
> Thank you for bringing this paper to our attention. There are indeed some apparent similarities: their Eq. 7 resembles our Eq. 5. Both involve minimizing divergences across a sequence of distributions.
> There are also some differences; most importantly, they are concerned with learning a generative model, whereas we learn a discriminative model. Also, it seems to us that the concept of "next step" captures something quite different in both problems.
>
> We are not well-versed enough in the nested importance sampling literature to be able to relate our work to theirs, beyond noticing that the optimization objective appears superficially similar. It seems to us that the two papers solve fundamentally different problems, but we do not exclude the possibility that there are deeper links between these two lines of research.
>
> > What kinds of confidence intervals did you use?
>
> We use 95% confidence intervals based on the standard error of the mean (`+- 1.96 * stderr`).
>
> > Figure 5 left - how come 1 token TC outperforms? How is it different from DCE if there's only 1 token? Shouldn't we expect consistency to make more of a difference with more generated tokens? I'm probably missing something here. Is it that you still generate the whole set of tokens, but only evaluate on a cut off set using labels from the whole set?
>
> Your last point is correct: we evaluate the model’s ability to discriminate between correct and incorrect generations given only the first $t$ tokens, but the labels come from the _full_ generation.
>
> Your questions raise two points.
>
> 1. _When should we expect TC-$\lambda$ to outperform DCE, with few tokens (i.e., short prefixes), or with a full completion?_ We expect short prefixes to benefit most from temporal-consistency, because that is when the soft targets in Eqs 5 & 7 are most dissimilar from the hard targets used in DCE. In fact, the loss term associated with full sequences is identical for TC-$\lambda$ and DCE (this can be seen by comparing Eq. 2 and the first term in Eq. 5). We were somewhat (pleasantly!) surprised to observe that TC-$\lambda$ models can also outperform DCE models on _full_ sequences.
> 2. _Setting aside differences between TC-$\lambda$ and DCE, how is it possible to achieve a non-trivial ROC AUC with only a single token?_ This is because, while the model can only see a single token from the generation, it can see the full problem statement. Even though the models are trained on a balanced dataset (every problem has one correct and one incorrect generation in the training data), the models seem to be able to recognize difficult problems statements at test time (and TC-$\lambda$ more so than DCE).

---

> > ### Comment · Reviewer_qrJB · 2025-08-01
> >
> > Thanks for your response!
> >
> > > We acknowledge that it is not at all easy to see that confidence intervals are present in the figures you mentioned, and we will experiment with alternative presentation styles (we also welcome your suggestions in that regard).
> >
> > One possibility is thicker bars to make them more visible. Alternatively, you could just include a note in the caption saying there are confidence intervals but they're so tight that they are basically invisible.
> >
> > > Thank you for bringing this paper to our attention. There are indeed some apparent similarities: their Eq. 7 resembles our Eq. 5. Both involve minimizing divergences across a sequence of distributions. There are also some differences; most importantly, they are concerned with learning a generative model, whereas we learn a discriminative model. Also, it seems to us that the concept of "next step" captures something quite different in both problems.
> > We are not well-versed enough in the nested importance sampling literature to be able to relate our work to theirs, beyond noticing that the optimization objective appears superficially similar. It seems to us that the two papers solve fundamentally different problems, but we do not exclude the possibility that there are deeper links between these two lines of research.
> >
> > I suspect there are deeper links, which is why I brought it up, but I'm not an expert with their work either. It might be just at a high-level, conceptual level; of course the application and the use cases are different.
> >
> > > When should we expect TC-$\lambda$ to outperform DCE, with few tokens (i.e., short prefixes), or with a full completion? We expect short prefixes to benefit most from temporal-consistency, because that is when the soft targets in Eqs 5 & 7 are most dissimilar from the hard targets used in DCE. In fact, the loss term associated with full sequences is identical for TC-$\lambda$ and DCE (this can be seen by comparing Eq. 2 and the first term in Eq. 5). We were somewhat (pleasantly!) surprised to observe that TC-$\lambda$ models can also outperform DCE models on full sequences.
> >
> > I agree that the final finding is surprising. Do you have insights (or guesses/ideas) on what is going on here?

---

> > > ### Author Response · Authors · 2025-08-04
> > >
> > > Thanks for your suggestions on error bars! We will try perpendicular caps & increasing width, and will default to a note in the caption if all else fails.
> > >
> > > Regarding insights for why TC-$\lambda$ models outperform DCE ones on full sequences, here are our thoughts.
> > >
> > > First, recall that for both TC-$\lambda$ and DCE, we observe that incremental classifiers outperform specialized classifiers trained on full sequences. Cobbe et al. [1] make a similar observation and note:
> > >
> > > >  We hypothesize that the full value function provides a useful auxiliary signal that encourages the model to judge the reasoning throughout solutions, rather than merely memorizing the correct final answer.
> > >
> > > We agree with their hypothesis: it seems that by forcing the model to make calibrated predictions on partial sequences, it is better able to learn which elements (e.g., which parts of the text) are relevant to accurately classifying the full sequence, helping it generalize better.
> > >
> > > By the same logic, one could imagine that by formulating a loss function that enables the model to make _even better_ predictions on partial sequences, we get a higher-quality "auxiliary signal" that further improves the performance on full-sequences.
> > >
> > > This could perhaps explain the improvement TC-$\lambda$ provides over DCE on full sequences.
> > >
> > > [1] K. Cobbe, V. Kosaraju, et al. Training verifiers to solve math word problems, arXiv:2110.14168.

---

> > > > ### Comment · Reviewer_qrJB · 2025-08-05
> > > >
> > > > Nice, thanks for your response and further discussion!

---

### Official Review · Reviewer_cjdV · 2025-07-02

**Clarity:** 3
**Significance:** 3
**Originality:** 3
**Rating:** 4
**Confidence:** 3

**Summary:**

This paper studies the problem of sequence classification of partial sentences. The method is related to temporal difference learning in reinforcement learning. The paper proves the convergence, consistency and data efficiency of the proposed approach. Over a number of datasets, the authors show classification accuracy improvement over baselines such as DCE, Last Token only baseline etc.

**Questions:**

- See the weakness part

- Why does the difference between the partial sentence classification accuracy (like 16 tokens) and all-token accuracy in Table 1 vary significantly across the datasets? Does the dataset affect the choices of hyperparameters like lambda?

- In Line 331: "scaling beyond than 1.3 B parameters remains an open question", this sentence is a bit confusing to me. Do you expect the proposed approach to face scalability issues under larger models (which I don't think so), or do you just mean you did not perform larger scale experiments due to computational constraints? Please enhance the clarity of the sentence.

**Ethical Concerns:**

["NO or VERY MINOR ethics concerns only"]

**Final Justification:**

Please update the writing as indicated in the response:
- better involve discussions of existing literature in the theoretical analysis part
- include more interpretations about the performance differences under different setups.


I don't have any more concerns about the paper.

**Limitations:**

Yes

**Quality:**

3

**Strengths And Weaknesses:**

**Strengths**

- Significance: the research problem is timely and important for LLMs that incrementally generate sequences by nature.

- Quality: The theoretical analysis of the consistency and data efficiency is solid. The analysis in Sec. 3.1 makes it convincing why the proposed TC outperforms DCE. The baselines are well-chosen, and the performance improvement is consistent over different model sizes and families.  The experiment section further shows consistent accuracy improvements under differently short partial sentences and the efficiency benefit of the approach. The two scenarios studied in the experiment section, text classification and LLM verification, are representative of the potential real-world application of the approach.

- Clarity: The paper is well organized and the writing is clear.

- Novelty: The novelty seems to lie on (1) applying the idea of TD learning to text classification over discrete label space (2) replacement of the squared loss to the cross entropy in the temporal consistency condition, which is a technically sound and natural choice. This novelty is acceptable for me.

**Weakness**

- Methodologically, the authors measures discrepancies with the cross entropy but conventional TD learning uses squared loss between the value function outputs. Adapting the TD learning idea to incremental sequence classification is non-trivial and interesting but I wonder whether the analysis like convergence of the approach is re-iteration (or minor adaptations) of known properties of TD learning. If so, citations are appreciated.

- The cited Mudgal et al. paper exemplifies a setup where the value function for continuous rewards, which perform verification of the generated text in a setup similar to Sec 4.2 of the paper. What is the advantage of the proposed approach compared to Mudgal et al. for verification of LLM generation verification? Will using the cross entropy loss perform better than the squared loss for LLM generation verification, for example, on the GSM8k studied?

- The experiment part can be strengthened - most of the points also mentioned in the limitations section by the author, like experiments in larger scale models (>1.3B parameters)

---

> ### Author Rebuttal · Authors · 2025-07-30
>
> Thank you for your review!
>
> > I wonder whether the analysis like convergence of the approach is re-iteration (or minor adaptations) of known properties of TD learning. If so, citations are appreciated.
>
> Indeed, our analysis builds on analogous properties of TD learning. We agree with your suggestion and plan to relate our results more systematically to the corresponding results for TD-learning. These appear in many forms across multiple references, but their presentation in [1] is perhaps the easiest to relate to our developments.
>
> [1] D. Bertsekas, J. Tsitsiklis. Neuro-Dynamic Programming. Athena Scientific, 1996.
>
> > Will using the cross entropy loss perform better than the squared loss for LLM generation verification, for example, on the GSM8k studied?
>
> This is a helpful comment, also raised by reviewer `mRRN`. Although the tasks we consider are naturally framed as classification problems with a binary or a categorical cross-entropy loss, they can indeed also be cast as regression problems with a squared loss. We have addressed this question in more details under `mRRN`'s review (see "Comparison to least squares methods"), including additional experiments comparing our approach to a standard TD-learning approach. We did not have time to run a comparison Qwen2.5 & GSM8K, but we hope that the experiments on the four text classification datasets of Section 4.1. provide some insight into this question.
>
> > Why does the difference between the partial sentence classification accuracy (like 16 tokens) and all-token accuracy in Table 1 vary significantly across the datasets?
>
> The datasets in Table 1 span a wide range of characteristics: some have around 20 classes (ohsumed, newsgroups) while others have only 2-4 classes (imdb, ag-news). The informativeness of the first 4 or 16 tokens also seems to vary significantly across datasets, independently of how hard the full-sequence classification task is. In that sense, we are not particularly surprised to see different patterns of performance across datasets.
>
> > Does the dataset affect the choices of hyperparameters like lambda?
>
> Yes, hyperparameters do change somewhat across datasets, including $\lambda$. The optimal values are given in Tables 3 & 4 in the appendix. The quantity $1 / (1 - \lambda)$ is the mean of the geometrically distributed weights in Eq. (7), assuming $T \to \infty$, and it has an intuitive interpretation as the _average lookahead_. In our experiments, the optimal average lookahead ranges between 5 and 50 tokens, depending on the dataset.
>
> Understanding what characteristics of the dataset correlate with optimal values $\lambda$ is an interesting question for future research.
>
> > In Line 331: "scaling beyond than 1.3 B parameters remains an open question", this sentence is a bit confusing to me. Do you expect the proposed approach to face scalability issues under larger models (which I don't think so), or do you just mean you did not perform larger scale experiments due to computational constraints? Please enhance the clarity of the sentence.
>
> We mean the latter (we did not perform larger-scale experiments due to computational constraints). Thank you for pointing this out, we will clarify this sentence in the next version of the manuscript.

---

> ### Comment · Reviewer_cjdV · 2025-08-05
>
> Thank you. Please update the writing as indicated in the response. I don't have any more concerns about the paper.

---

### Official Review · Reviewer_C7UX · 2025-07-02

**Clarity:** 4
**Significance:** 3
**Originality:** 3
**Rating:** 5
**Confidence:** 3

**Summary:**

The paper introduces a condition to express consistency between predictions in incremental sequence classification tasks and proposes a corresponding loss function to enforce this property during the training of a parametric model.

**Questions:**

1. Equation (5) resembles, to me, the typical form of a cost functional in optimal control problems, where the first term corresponds to a terminal cost and the second term resembles the integral of a running cost. However, in line 122, you remark that this equation can be seen as a form of the Bellman equation, suggesting a connection to value functions. Could you please elaborate on this?

2. You mention that the computational overhead introduced by the proposed method is negligible, as training time is nearly indistinguishable from the baseline. However, it would be helpful to quantify this claim more precisely, perhaps by providing a theoretical estimate or reporting empirical runtime differences in a controlled setting.

**Ethical Concerns:**

["NO or VERY MINOR ethics concerns only"]

**Final Justification:**

I have carefully read the authors’ rebuttal as well as the comments from the other reviewers and the AC. My questions were answered satisfactorily, and I did not find any additional major issues raised by the other reviewers. For these reasons, I maintain my positive score.

**Limitations:**

yes

**Quality:**

3

**Strengths And Weaknesses:**

The manuscript is extremely well written. The problems are clearly introduced and motivated, and all assumptions are adequately discussed.

The proposed method is analyzed both theoretically—demonstrating convergence and consistency—and experimentally, through evaluations on standard text classification benchmarks and a specific model verification task for LLMs. The experiments are well designed.

If I understand correctly, the theoretical analysis in Section 3 relies heavily on the assumption of working with tabular data, which does not hold for the experiments presented in Section 4. While one might hope that some desirable properties carry over to the parametric setting, the manuscript does not provide strong reasons or intuitions for why this should be the case.

In terms of novelty, the idea of enforcing temporal consistency across successive prefixes in a sequence classification task is a meaningful extension of related principles from reinforcement learning into a multiclass classification setting. This, to the best of my knowledge, is an original contribution.

This work advances the field from a theoretical point of view  by introducing a new temporal-consistency loss. The method is also proven consistent in a tabular setting and shown empirically to improve predictive performance.

---

> ### Author Rebuttal · Authors · 2025-07-30
>
> Thank you for your review!
>
> > If I understand correctly, the theoretical analysis in Section 3 relies heavily on the assumption of working with tabular data, which does not hold for the experiments presented in Section 4. While one might hope that some desirable properties carry over to the parametric setting, the manuscript does not provide strong reasons or intuitions for why this should be the case
>
> We acknowledge that Section 3.1 does not provide much intuition about what makes our temporal-consistency loss particularly effective on the problem in Figure 1, and how this intuition might carry over in more realistic applications with parametric models. We hope to be able to improve this in the next version of the manuscript, given some additional space.
>
> Informally, temporally-consistent methods (such as TC-$\lambda$) exploit information from the dynamics of the sequence. For example, in text classification, one could argue that there are many different ways of organizing words into sentences that capture a similar “semantic state” that is predictive of the class label. In this case, encouraging predictions to be consistent across successive states improves data-efficiency: the model is now implicitly leveraging information from sequences that might have started very differently but reached the same intermediate semantic state. (This data-pooling phenomenon is underpinning the improved statistical efficiency of TC-$\lambda$ over DCE on the problem of Sec. 3.1)
>
> > Equation (5) resembles, to me, the typical form of a cost functional in optimal control problems, where the first term corresponds to a terminal cost and the second term resembles the integral of a running cost. However, in line 122, you remark that this equation can be seen as a form of the Bellman equation, suggesting a connection to value functions. Could you please elaborate on this?
>
> In our setup, $p_\theta(\cdot \mid s_t)$ plays the role of the value model. Consider the left-hand side (LHS) and right-hand side (RHS) of Equation (4). Equation (5) can be understood as penalizing the KL-divergence of the LHS relative to the RHS, under an empirical Markov model $p(s_{t+1} \mid s_t)$ given by the data. (Proposition 2 in Sec. 3 makes this precise.)
>
> It is perhaps helpful to make an analogy to a value model in a Markov reward process with discount factor $\gamma$. In that case, the Bellman equation is
>
> $$
> V(s_t) = \mathbb{E}  \quad_{p(r_{t+1}, s_{t+1} \mid s_t)} [ r_{t+1} + \gamma V(s_{t+1})]
> $$
>
> And the analogue to our Equation (5) would penalize the squared Bellman error, i.e., the squared difference between the LHS and RHS as
>
> $$
> \sum_{t=1}^T \sum_{(s_t, r_{t+1}, s_{t+1})} [V_{\theta}(s_t) - (r_{t+1} - \gamma V_\theta(s_{t+1}))]^2
> $$
>
> > You mention that the computational overhead introduced by the proposed method is negligible, as training time is nearly indistinguishable from the baseline. However, it would be helpful to quantify this claim more precisely, perhaps by providing a theoretical estimate or reporting empirical runtime differences in a controlled setting.
>
> To answer your question, we have re-analyzed a sample of our training runs’ logs. Empirically, the time per training step of TC-$\lambda$ appears to be within 1% of that of DCE. This is explained by the fact that the only overhead of TC-$\lambda$ over DCE is the computation of the pseudo-targets $\{z_t\}$ in (7). This is several orders of magnitude faster than computing forward and backward passes of an LLM. We thank you for your question and will clarify this in the manuscript.

---

> > ### Comment · Reviewer_C7UX · 2025-08-08
> >
> > Thank you for your responses. As you saw from my score, I already had no major concerns about the paper.

---

### Official Review · Reviewer_mRRN · 2025-07-13

**Clarity:** 3
**Significance:** 3
**Originality:** 3
**Rating:** 5
**Confidence:** 2

**Summary:**

This paper propose a new loss function, TC-\lambda, for training incremental sequence classifiers. The authors is inspired by TD learning and propose a method that enforces consistency between prediction distribution between different timestamps. The paper provides some theoretical analysis as well as empirical results on text classification and large language model (LLM) verification tasks.

**Questions:**

- Is there attempt to anneal the \lambda throughout the training process?

**Ethical Concerns:**

["NO or VERY MINOR ethics concerns only"]

**Final Justification:**

The authors well addressed my question on lambda annealing and difference compared to TD-lambda.
I remain my original rating.

**Limitations:**

yes

**Paper Formatting Concerns:**

No.

**Quality:**

3

**Strengths And Weaknesses:**

Strength:
- The method is well motivated. The connection between TC-lambda and TD learning is straightforward and easy to intuitive.
- The authors provide comprehensive empirical evaluation on text classification and LLM verification.

Weaknesses:
- Given that the model can do verification task, it should be able to serve as a good value function. It would be great if the authors actually use the value function to finetune a LLM.
- My understanding is the authors originally want to learn an incremental value function for training LLM using RL. So how is TC-lambda different from simply train a value function using TD learning? One might simply use the original TD learning to train such an incremental sequence classification task, but there is no comparison on that in the paper.

---

> ### Author Rebuttal · Authors · 2025-07-30
>
> Thank you for your review!
>
> > how is TC-lambda different from simply train a value function using TD learning? One might simply use the original TD learning to train such an incremental sequence classification task
>
> This is a helpful comment, also raised by reviewer `cjdV`. Although all the tasks we consider are naturally framed as classification problems with a binary or a categorical cross-entropy loss, they can indeed also be cast as regression problems with a squared loss. We address this question below in more details, including additional experiments comparing our approach to a standard TD-learning approach.
>
> > Is there attempt to anneal the \lambda throughout the training process?
>
> We have not directly experimented with continuous annealing schedules, but we have noticed that the following two-step process (which can be considered as a basic form of annealing) works well:
>
> 1. Initialize the classification head by training a linear probe with $\lambda = 1$.
> 2. Unfreeze the language model parameters and fine-tune end-to-end with the desired value of $\lambda$.
>
> While not strictly necessary to achieve good performance, we have noticed that this leads to very stable and robust training runs that converge quickly. This is the approach we have adopted for our experiments.
>
> Continuously annealing lambda during step 2 above is an interesting idea for future work.
>
> # Comparison to least squares methods
>
> In our paper, we train classifiers with a softmax cross-entropy loss.
> Alternatively, we could frame the classification task as a regression problem (predicting a final scalar reward, which in our case is either $0$ or $1$; in the multiclass case, the $K$ classes can be modeled independently).
> This regression-based framing naturally suggests using standard value-estimation algorithms from the RL literature, which use a squared loss.
>
> With this in mind, we ran two additional baselines on the four text classification datasets:
>
> - **Direct squared loss**: regress the final reward directly. This the squared-loss variant of DCE.
> - **LSTD-$\lambda$**: use temporal-difference learning, specifically the offline version of [1]. This is the squared-loss variant of TC-$\lambda$.
>
> [1]: S. Bradtke, A. Barto. Linear Least-Squares Algorithms for Temporal Difference Learning. Machine Learning, 1996.
>
> We compare the predictive performance of these regression baselines on the four text classification datasets of Section 4.1 below.
>
> Key:
>
> - **oh**: ohsumed
> - **ng**: newsgroups
> - **im**: imdb
> - **ag**: ag-news
>
> Accuracy (higher is better):
>
> |                     | oh-4 | oh-16 | oh-full | ng-4 | ng-16 | ng-full | im-4 | im-16 | im-full | ag-4 | ag-16 | ag-full |
> | ------------------- | ---- | ----- | ------- | ---- | ----- | ------- | ---- | ----- | ------- | ---- | ----- | ------- |
> | DCE                 | 30.5 |  65.5 |    81.1 | 27.7 |  40.1 |    **89.0** | 63.5 |  74.7 |    94.4 | 80.0 |  92.6 |    94.8 |
> | TC-$\lambda$        | **33.7** | **68.3** | **81.8** | **33.4** | **44.7** | 88.5 | **64.7** | **75.7** | **94.9** | **81.4** | **93.0** | **95.0** |
> | Direct squared loss | 30.3 |  65.0 |    80.4 | 19.7 |  29.9 |    88.3 | 63.6 |  74.7 |    94.3 | 80.0 |  92.6 |    94.7 |
> | LSTD-$\lambda$      | 32.7 |  64.9 |    78.0 | 26.2 |  36.7 |    87.8 | 64.6 |  75.4 |    94.7 | 81.1 |  92.8 |    94.9 |
>
> Negative log-likelihood (lower is better):
>
> |                     | oh-4 | oh-16 | oh-full | ng-4 | ng-16 | ng-full | im-4 | im-16 | im-full | ag-4 | ag-16 | ag-full |
> | ------------------- | ---- | ----- | ------- | ---- | ----- | ------- | ---- | ----- | ------- | ---- | ----- | ------- |
> | DCE                 | 2.428 | 1.241 |  0.663 | 2.783 | **2.538** |  0.616 | 0.626 | 0.497 |  0.155 | 0.545 | 0.226 |  0.158 |
> | TC-$\lambda$        | **2.365** | **1.162** |  **0.646** | **2.777** | 2.635 |  **0.503** | **0.621** | **0.484** |  **0.138** | **0.510** | **0.209** |  **0.148** |
> | Direct squared loss | 2.964 | 2.636 |  2.485 | 2.845 | 2.741 |  2.209 | 0.632 | 0.545 |  0.369 | 0.958 | 0.829 |  0.804 |
> | LSTD-$\lambda$      | 2.993 | 2.649 |  2.454 | 2.907 | 2.817 |  2.192 | 0.638 | 0.553 |  0.372 | 0.966 | 0.825 |  0.799 |
>
> Take-aways:
>
> - methods based on the softmax cross-entropy loss outperform methods based on squared-loss
>     - in terms of accuracy, the difference is relatively small
>     - the predictive log-likelihood, however, is very poor for regression models (as can be expected).
> - as could be expected, the difference is larger for problems with many classes (ohsumed, newsgroups)
> - temporally-consistent methods (TC-$\lambda$, LSTD-$\lambda$) tend outperform direct methods (DCE, direct squared loss), especially for short prefixes.
>
> In a way, TC-$\lambda$ provides the best of both worlds: it leverages the benefits of temporal-difference learning while optimizing a loss function that is arguably better suited for classification.

---

### Decision · Program_Chairs · 2025-09-17

**Decision:**

Accept (spotlight)

**Comment:**

The paper proposes an algorithm for incremental sequential classification, where predictions are progressively updated as new elements in the sequence are revealed, guided by a temporal-consistency condition.

Based on reviewers’ comments, the paper demonstrates the following strengths:
* The problem is well-motivated, and the paper is clearly written.
*  Enforcing temporal consistency in sequential classification is novel, and the proposed temporal-consistency loss represents a new contribution.
* The empirical evaluation is well-designed and comprehensive, with competitive baselines and consistent performance improvements.

Reviewers raised questions regarding comparisons with original TD learning, analysis of computational overhead, and comparison with Nested Variational Inference etc.. Most of these concerns were addressed in the rebuttal. The authors are encouraged to incorporate the additional results and suggestions from reviewers into the final version to further strengthen the work.